# Towards a harmonized operational earthquake forecasting model for Europe

**Marta Han, Leila Mizrahi, and Stefan Wiemer**

Swiss Seismological Service (SED), ETH Zurich, Zurich, Switzerland

**Correspondence:** Marta Han (marta.han@sed.ethz.ch)

**Abstract.** We develop a harmonized earthquake forecasting model for Europe based on the epidemic-type aftershock sequence (ETAS) model to describe the spatiotemporal evolution of seismicity. We propose a method modification that integrates information from the 2020 European Seismic Hazard Model (ESHM20) about the spatial variation in background seismicity during ETAS parameter inversion based on the expectation–maximization (EM) algorithm. Other modifications to the basic ETAS model are explored, namely fixing the productivity term to a higher value to balance the more productive triggering by high-magnitude events with their much rarer occurrence and replacing the $b$-value estimate with one relying on the $b$-positive method to observe the possible effect of short-term incompleteness on model parameters. Retrospective and pseudo-prospective tests demonstrate that ETAS-based models outperform the time-independent benchmark model as well as an ETAS model calibrated on global data. The background-informed ETAS variants using the $b$-positive estimate achieve the best scores overall, passing the consistency tests and having a good score in the pseudo-prospective experiment. Our test results also highlight promising areas for future exploration, such as avoiding the simplification of using a single $b$ value for the entire region, reevaluating the completeness of the used seismic catalogs, and applying more sophisticated aftershock spatial kernels.

## 1 Introduction

After the occurrence of a high-magnitude earthquake, the expected behavior of its aftershocks and, in particular, the possibility of another large event are of interest to both the general public and governmental and private organizations, such as civil protection services, first responders, and insurance companies. Operational earthquake forecasting (OEF; Jordan et al., 2011) was introduced as a term for "gathering and disseminating authoritative information about the time dependence of seismic hazards to help communities prepare for potentially destructive earthquakes". It is an evolving effort that has seen significant progress in recent years. Several countries, including New Zealand (Christophersen et al., 2017), the United States (Field et al., 2017; Jordan et al., 2011, 2014; van der Elst et al., 2022), and Italy (Marzocchi and Lombardi, 2009; Marzocchi et al., 2014) currently have systems in place that produce authoritative earthquake forecasts (see Mizrahi et al., 2024b, for a review of these OEF systems and Hardebeck et al., 2024, for a review on aftershock forecasting). Each of these systems uses different underlying models to produce the forecasts and communicates the forecasts differently (absolute or relative earthquake rates/probabilities, maps, scenarios, etc.) and to different user groups at different time intervals (continuously, daily, monthly, or only after large events), and the forecasting systems are continuously being modified and improved. In short, there is not a unique agreed-upon best way to provide OEF, as has been shown recently in an elicitation of expert views on this topic (Mizrahi et al., 2024b).

However, what is clear is that to issue earthquake forecasts operationally, time-dependent models that describe both the spatial and temporal variability in seismicity are required. Gathering historic and recent seismicity data, combined with knowledge of seismotectonic properties of a region, allows us to better understand spatial variability in earthquake occurrence in a time-independent manner (Danciu et al., 2021; Crowley et al., 2021; Wiemer et al., 2016). In addition to

the time-independent assessment of seismicity, the temporal evolution of seismic sequences can be modeled using well-established empirical laws, as has been done by several (governmental or non-governmental) agencies on various scales (Christophersen et al., 2017; Field et al., 2017; Jordan et al., 2011; Mizrahi et al., 2024a; Marzocchi and Lombardi, 2009; Nandan et al., 2021; Omi et al., 2018).

The main objective of this paper is to develop a harmonized forecasting model for Europe that represents the current state of the art of time-dependent earthquake forecasting. The idea of a harmonized model is to take into account not only the differences in data collection properties but also physical properties of various tectonic regions to minimize the effect of administrative borders on the output, providing a unique set of parameters that, in a way, average seismicity properties in the observed region, hopefully benefiting from the variety of events present in such a large dataset. This model is meant to be simple and serve as the basis for the development of future models and is not meant to overrule other, national forecasting models where they are available (e.g., Italy; Marzocchi et al., 2014). We aim to identify the shortcomings of a basic harmonized model and propose modifications that would remediate them with the goal of providing reliable earthquake probabilities incorporating long-term seismicity rates as well as short-term clustering patterns.

The current state-of-the-art models for time-dependent earthquake forecasting are epidemic-type aftershock sequence (ETAS) models (Ogata, 1988). In the context of ETAS models, any event triggered by its predecessor is named an aftershock, which is not to be confused with the frequent interpretation of the term aftershock meaning an event of smaller magnitude following its triggering event. Having been introduced by Ogata in 1988, these models have been around for several decades, implemented and used by many agencies, and identified by experts in the study of Mizrahi et al. (2024b) as the preferred choice for a default model to be used for earthquake forecasting. Their main strength is in explaining the aftershock-triggering behavior of earthquakes, relying on the temporal and spatial decay of the number of aftershocks with spatial or temporal distance from the main event, the productivity law, and the Gutenberg–Richter (GR) law (Omori, 1900; Utsu, 1971; Gutenberg and Richter, 1936). In ETAS, the seismicity rate $\ell$ is given as the sum of the background rate $\mu$ and the aftershock rate $g$ of all previous events, following these laws. Specifically, we will use the formulation as in Nandan et al. (2021) and Mizrahi et al. (2021b):

$$\ell(t, x, y) = \mu + \sum_{i:t_i < t} g(m_i, t - t_i, x - x_i, y - y_i),$$

$$g(m_i, t - t_i, x - x_i, y - y_i)$$
$$= \frac{e^{-(t-t_i)/\tau} \cdot k_0 e^{a(m_i - m_c)}}{(t - t_i + c)^{1+\omega}((x - x_i)^2 + (y - y_i)^2 + de^{\gamma(m - m_c)})^{1+\rho}}.$$
$$\tag{1}$$

Training such a model for the European region poses a number of challenges, as has been laid out by Zechar et al. (2016). A main challenge lies in the compilation of a dataset containing a comprehensive record of earthquakes over a significant period of time, especially considering the differing formats and properties used in different countries or subregions (e.g., magnitude types and their definitions, data completeness due to network density and other reasons, location and magnitude precision). Moreover, it is desirable to leverage high-quality data (higher precision, completeness to a lower magnitude) where available, without losing potentially valuable information about high-magnitude events in periods and regions wherein data collection was not as precise and complete. Data completeness is often quantitatively expressed through the completeness magnitude ($m_c$), which is the lowest magnitude above which all events are assumed to be observed. A catalog of all recorded events is normally incomplete, meaning that it also contains events below $m_c$, and as the exact $m_c$ is not known, it is important to estimate $m_c$ and remove events below $m_c$. Underestimating it may bias models trained on the data with an $m_c$ higher than assumed (Seif et al., 2017), but overestimating it results in throwing away complete and potentially useful data.

Multiple methods for estimating $m_c$ have been developed and tested, mostly relying on the fact that, by the Gutenberg–Richter law, the events in a complete catalog follow an exponential distribution, with their cumulative count satisfying

$$N(m) = 10^{a-bm}, \tag{2}$$

where $N(m)$ denotes the number of events with a magnitude of $m$ and above and $a$ and $b$ are parameters often referred to as the $a$ and $b$ value. Note that the natural logarithm base is also used, in which case

$$N(m) = N_0 e^{-\beta m}, \quad N_0 = 10^a, \quad \beta = b \ln 10. \tag{3}$$

Although challenging, recent achievements in data collection and harmonization have enabled the creation of a Europe-wide earthquake catalog which we aim to use as a basis for the calibration of a Europe-wide ETAS model in this study. The main result that will be used in this study in terms of data gathering is the catalog collected for the 2020 European Seismic Hazard Model (ESHM20; Danciu et al., 2021), which provides harmonized information about seismic activity on an overall European scale, relying on expert knowledge about the differences in earthquake monitoring and physical tectonic characteristics of the region in order to harmonize the data and providing elicitation both on data properties (such as $m_c$) and division into subregions based on their seismotectonic properties. Moreover, the ability to fit ETAS models to datasets with varying $m_c$ values, not only originally developed for time-varying $m_c$ but also applicable for any spatial variations in completeness, introduced by Mizrahi et al. (2021b) allows for ETAS models to use the

high-quality data in both (more recent) time periods and sub-regions with low $m_c$ values and potentially capture long-term trends contained in periods and areas with higher $m_c$ values.

Besides a basic ETAS model, we will consider several modifications and test them both retrospectively for self-consistency and pseudo-prospectively for comparison against one another. While the main strength of ETAS models is in modeling aftershock behavior, it is expected that the background rate varies significantly in space over a large area such as Europe. One of our main proposed modifications focuses on implementing the knowledge about spatial variations in the background rate inferred by ESHM20 (Danciu et al., 2021) already during the inversion of ETAS parameters, which could affect the parameters describing aftershock behavior as well. These spatially varying background seismicity rates are estimated leveraging both the area source model and the background seismicity and active-faults model from ESHM20, combined with equal weighting as proposed by Danciu et al. (2021). Other modifications include fixing the term dictating the productivity law to the $b$ value of the catalog (Hainzl et al., 2013; van der Elst et al., 2022) to balance the more productive triggering by high-magnitude events (productivity law; Utsu, 1971) with their much rarer occurrence (Gutenberg–Richter magnitude distribution law; Gutenberg and Richter, 1936) as explained in Helmstetter (2003) and implementing the $b$-positive method (van der Elst, 2021) for the estimation of the $b$ value.

The outline of the remaining sections of this article is the following: in Sect. 2, we briefly describe the ESHM20 catalog with both its more recent and its historic parts and then describe the selection of the time frame used in this study due to computational limitations and high heterogeneity in data quality between time periods. We introduce additional data about long-term seismicity given by ESHM20 that will be used as input to some model variants and the most recent catalog that will be used for model validation. The development of a base model and modifications thereof are described in more detail in Sect. 3, followed by a description of the methods used for testing them (Sect. 3.3 and 3.4). Finally, our results are presented and discussed in Sect. 4, divided into three parts, presenting the fitted parameters of the models, results of retrospective consistency tests, and results of pseudo-prospective model comparison experiments.

## 2 Data

The primary dataset used in this work is the ESHM20 catalog (Danciu et al., 2021), which contains the combined catalogs of all agencies that record earthquakes in Europe, both recent and historical, dating back to the 11th century. Due to the variations in both the nature of earthquake occurrence and its monitoring, the data are highly heterogeneous. For pre-instrumental times, the records are highly incomplete, potentially even missing high-magnitude events and containing errors higher than 0.5 magnitude units for the magnitudes of the events that are in the record (Grünthal et al., 2009; Grünthal and Wahlström, 2012). Constructing the catalog focusing on the period starting with the 20th century already involved parameterizing earthquakes from macroseismic data mixed with instrumentally recorded events compiled in 47 subregions and harmonizing the magnitudes calibrated in scales such as local magnitude, body-wave magnitude, surface-wave magnitude, moment magnitude, and maximum intensity into one equivalent to the moment magnitude ($M_W$). This follows the methodology of Grünthal and Wahlström (2012), applying a hierarchical strategy which prioritizes existing $M_W$-harmonized catalogs, followed by moment tensor databases and local bulletins and finally data from the International Seismological Centre (ISC) when no local data are available. Magnitude conversions also follow Grünthal and Wahlström (2012), with updates from recent literature where applicable.

As the density and sensitivity of seismic networks generally improve over time, the magnitude and location precision increases (Danciu et al., 2021), as well as the number of recorded events due to the ability to record lower-magnitude events (the completeness magnitude $m_c$ decreases). However, neither the level of completeness nor this improvement over time is spatially uniform. Between some regions, in the same period, the $m_c$ difference can be up to 4 magnitude units. Although a more precise magnitude resolution is available for part of the data, assuming a higher precision when it is not available in parts of the catalog would produce incorrect estimates. The agreed-on precision is 0.2, as in Danciu et al. (2021), used there also for the statistical fitting of the seismicity parameters of the source models.

Due to computational limitations, poor quality, and incompleteness of early data, making it unsuitable for the analysis of aftershock behavior, the dataset needs to be narrowed down to contain relatively recent information while ensuring a sufficiently long time frame that enables the capturing of longer-term triggering effects and seismicity patterns. High-magnitude events that are present in historical parts of the catalog are crucial to better understand the frequency of rare seismic events that are not present in more recent time periods, and they help to identify additional spatial patterns in background seismicity. However, these historical high-magnitude events will seldom have aftershocks recorded due to the incompleteness reflected in $m_c$ going as high as a magnitude of 8. According to the completeness magnitudes for different time periods and regions assessed by experts in ESHM20 (Danciu et al., 2021) and visualized in Fig. S1 in the Supplement, up to the early 1980s, the highest completeness difference between regions in the same time period is as high as 3 magnitude units; starting from the 1990s, this difference lowers to 1.5 units of magnitude. Hence, we here limit the catalog to the time period starting with the year 1980, with only events after 1990 considered to be potentially triggered events (this is discussed in more detail later,

in Sect. 3.2.1). The spatial distribution of the catalog containing over 20 000 events in this time period is shown in Fig. 1a (in red). While the aforementioned issues of earthquake monitoring in earlier parts of the catalogs are fewer in the selected recent time period, the effect of neglecting to address them (such as assuming too low an $m_c$ or too high a magnitude precision) could still potentially significantly bias our later output (Seif et al., 2017).

By definition of the completeness magnitude, all events of magnitude equal to or higher than $m_c$ are assumed to be recorded. The events of magnitude below $m_c$ are also present in the raw catalog, resulting in incompleteness in the data. The incompleteness in this dataset below magnitude 4.6 is so evident that it is already detectable through visual inspection of Fig. 1b–c. Namely, under the assumption that the observed number of events does not have a significant trend over a longer period of time, the cumulative count of events through time would display roughly linear increasing behavior, with rapid jumps in the count at a point indicating only the occurrence of a productive sequence of events. The changes in the slope of this linear increase shown in (c) indicate the changing completeness over that time period and an increase in the rate of the cumulative earthquake count, suggesting that completeness improves over time. In (b), both incompleteness and discretization are discernible in the plot showing recorded magnitudes over time.

As mentioned earlier, the study of Danciu et al. (2021) provides expert evaluations of the completeness magnitudes by region and time period. Knowing these $m_c$ values allows for accounting for the incompleteness of data later during model calibration as described in Mizrahi et al. (2021b). As $m_c$ differs between regions and time periods, the distribution of $m - m_c(x, y, t)$ is shown in Fig. 1d instead of a distribution of "pure" magnitudes to correct each magnitude for the corresponding incompleteness level. Lines representing $b$ values of 1.23 and 0.99 are shown, as these are the estimates we use later in the calibration of ETAS models and simulation of synthetic catalogs, with former being computed as a "standard" maximum likelihood estimation with the binning correction suggested by Tinti and Mulargia (1987) and the latter being the $b$-positive (van der Elst, 2021) estimate, also with binning correction. The standard deviation estimated by the method of Shi and Bolt (1982) of the $b$-value estimates is 0.015 for the "traditional" and 0.036 for the $b$-positive method.

The catalog of Danciu et al. (2021) ends in 2015, and we use it in full for model training. The continuation of the catalog is given in Lammers et al. (2023) until 2022, and this 7-year period is used here for pseudo-prospective testing. This new part of the catalog is not identical in composition to the catalog used for training, with the most prominent difference being the completeness magnitude of 4.6 in the overall dataset (demonstrated in Fig. 1b). The spatial distribution of the testing catalog is shown in Fig. 1a (in green). In truly prospective testing, such differences in both compo-

sition methods and the content of the catalogs not only are possible but rather are an expected occurrence, the effect of which is not to be disregarded but rather leveraged to obtain more robust models. While this would not necessarily happen in a network controlled by a single agency, except for improvements in completeness, our catalog is composed of subregions and tremendous effort has been given, as described earlier, to harmonize it, and thus near-real-time deployment would possibly need to be replaced by an alternative for completeness and other properties. Note that in both the training and the testing catalog, due to the binning of $\Delta m = 0.2$ mentioned above, a completeness magnitude $m_c$ means that it actually contains events above $m_c - \frac{1}{2}\Delta m = m_c - 0.1$.

Furthermore, in this study, alongside earthquake catalogs, we aim to utilize the long-term seismicity rates introduced by Danciu et al. (2021). These rates are provided for both the area source model and the background seismicity and active-faults model. The area source model is a classical seismogenic source model, describing seismicity as shallow crustal, volcanic, in-slab subduction and deep, relying on recommendations by regional and national experts with modifications made to ensure compatibility in bordering (overlapping) areas. The background seismicity and active-faults model combines the smoothed background seismicity model obtained by estimating activity parameters ($a$ and $b$ value in the GR law) for a declustered complete catalog and the model describing seismic productivity in the proximity of faults with a fault-dependent magnitude threshold between them ensuring avoiding double counting seismicity. As in ESHM20, the annual seismicity rates for each spatial and magnitude bin are obtained by combining the outputs of these two models, with equal weighting. Summing the rates across all magnitude bins and accounting for differences in the completeness magnitude and time duration yields overall daily background seismicity rates for the spatial bins defined in the study. The final rates per spatial bin are visualized in Fig. S2. Although these rates are based on declustered seismicity, which should closer correspond to the background rate in ETAS as aftershock clustering has been removed, we want our models to invert the overall background rate freely, therefore only using this information as an input for relative spatial differences in the background rate. By adding this extra input, we include information from the historical periods of the ESHM20 catalog for large events and seismicity in areas not represented in the selected training part of the catalog (after 1980).

## 3 Methods

### 3.1 ETAS

Training an ETAS model on a given dataset means finding the parameters in Eq. (1) that give the best fit to the data. The inversion of the ETAS parameters $\mu$, $k_0$, $a$, $c$, $\omega$, $\tau$, $d$, $\rho$ and $\gamma$ used here is based on an expectation–maximization (EM)

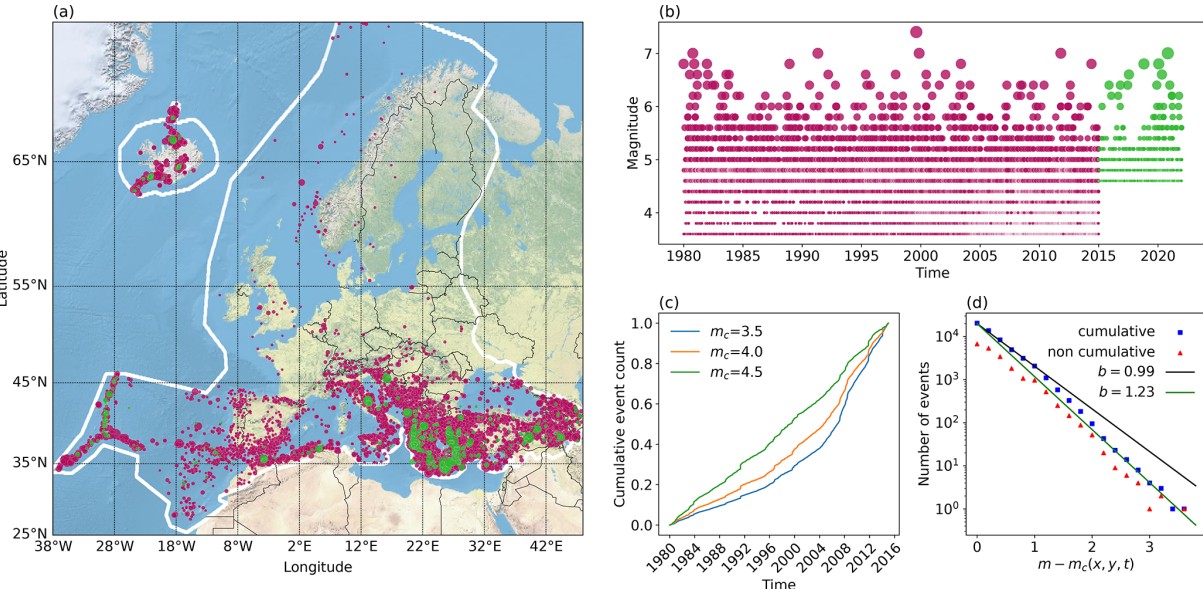

**Figure 1.** Dataset used for model training (1980–2015, red dots) and testing (2015–2022, green dots); in **(c)**–**(d)** only the training dataset is shown. **(a)** Map of events detected in the study area defined in ESHM20, which is outlined in white. The dot size increases with magnitude. **(b)** Time evolution of recorded events' magnitudes. The dot size increases with magnitude. **(c)** Cumulative count of recorded events through time for different cutoff magnitudes. **(d)** Magnitude frequency distribution plot. The distribution of $m - m_c(x, y, t)$ is shown to correct for varying levels of completeness. The lines show the fits of $b$ values estimated by the standard maximum likelihood method ($b = 0.99$, $\sigma = 0.015$) and $b$-positive method ($b = 1.23$, $\sigma = 0.036$).

algorithm (Veen and Schoenberg, 2008), with the varying $m_c$ adjustment (Mizrahi et al., 2021b). Conservatively using the maximum value of $m_c$ across the entire catalog would result in the loss of a large amount of valuable data, while assuming a completeness magnitude lower than the true one could introduce biases into our calculations (Seif et al., 2017).

In this modified EM algorithm by Mizrahi et al. (2021b), the difference between the overall lowest completeness magnitude $m_{ref}$ and the completeness magnitude at the location and time of a given event $m_c(x, y, t)$ is taken into account for each event by estimating the ratio of the unobserved and observed events ($\zeta$) and the ratio of events triggered by unobserved and observed events ($\xi$) based on the Gutenberg–Richter magnitude distribution assumption. The algorithm was implemented in Python by Mizrahi et al. (2023) and can, in principle, be used to calibrate basic ETAS models for any given catalog.

As the computation of $\zeta$ and $\xi$ relies on the GR law to estimate the number of unobserved events, this method is dependent on the estimated $b$ value of the catalog. Therefore, we test both the classical maximum likelihood method and the $b$-positive method (van der Elst, 2021), which is meant to overcome incompleteness in data, primarily the short-term aftershock incompleteness (STAI; Kagan, 2004). In both cases, we adjust for magnitude binning as described in Tinti and Mulargia (1987), which is especially important to avoid biases in the $b$ value due to the relatively large bins of $\Delta m = 0.2$.

In an ETAS model which produces a unique set of parameters for the overall region, information and properties of specific faults and sequences are potentially lost, but the advantage of global models is in their training datasets containing a larger number of high-magnitude earthquakes (Bayona et al., 2023). Another approach for mitigating the averaging behavior of a global model is to update the aftershock behavior described by ETAS with real-time data from an ongoing sequence when operationally issuing aftershock forecasts (Omi et al., 2015; van der Elst et al., 2022). The implementation of such a sequence-specific model updating, however, still involves a number of expert decisions, and there is no agreed-upon updating approach that yields the best overall performance of a global model. For example, the OEF systems of Italy, New Zealand, and the United States all use different approaches for updating their model parameters (Mizrahi et al., 2024b). In Italy, individual models are not updated, but their weights in the model ensemble are determined based on their past performance (Marzocchi et al., 2012). In New Zealand, ETAS parameters were fitted using data prior to 2012 (Harte, 2013) and have not been updated since. The USGS applies sequence-specific forecasts, but applying one strategy for all sequences increases performance in some cases but decreases it in others (van der Elst et al., 2022). While this is a feasible direction for future European model development, here we focus on developing a baseline, harmonized model upon which such improvements could be built.

## 3.2   Model variants

In addition to fitting an overall generic ETAS model to our dataset, in this section we propose modifications that could, in principle, be applied to any ETAS model. The models compared in this study are as follows.

– $ETAS_0$. This basic ETAS model is a set of parameters fitted to the ESHM20 dataset with no additional input or constraints. The implementation relies on the EM algorithm (Veen and Schoenberg, 2008) with varying $m_c$ modifications (Mizrahi et al., 2021b). The background rate is modeled by a single parameter during EM parameter inversion, but more background events are simulated in locations where background seismicity was observed in the training catalog.

– $ETAS_{bg}$. In order to be consistent with the long-term model (ESHM20; Danciu et al., 2021) and to utilize the information contained in the hazard model about spatially varying seismicity rates, the parameter inversion algorithm is modified to allow for variations in the background rate, keeping relative spatial information fixed.

– $ETAS_\alpha$. Due to the observed behavior of ETAS models to underestimate the productivity of high-magnitude events, the parameter dictating the productivity law $\alpha = a - \rho\gamma$ is fixed to $\alpha = \beta$ as a constraint during inversion.

– $ETAS_{bg,\alpha}$. The two proposed modifications are combined.

– $ETAS_0^{b+}$, $ETAS_{bg}^{b+}$, $ETAS_\alpha^{b+}$, $ETAS_{bg,\alpha}^{b+}$. These are the four model variants introduced above, with the only difference being in the $b$-value estimation method (van der Elst, 2021).

– $ETAS_{USGS}$. To add a comparison level and check for the benefits of fitting an ETAS model specific to European data, we use the parameters from the prior models described in van der Elst et al. (2022), applied by the USGS AftershockForecaster software. This includes several simplifications and adjustments, namely, background seismicity and the aftershock spatial kernel is taken from $ETAS_0$ and the global average is considered for other parameters.

– *Poisson background model*. We implement a time-independent model that takes the seismicity rate map provided by ESHM20 (Danciu et al., 2021) and, for each spatial cell in this map, forecasts a number of events following the Poissonian distribution with the corresponding rate in that cell as a mean. This is the null model against which comparisons are made in the testing phase to check for the performance of added time-dependent information during aftershock sequences. The rate map by ESHM20 only considers data up to 2015 and contains no information from the later time period set aside for pseudo-prospective testing.

### 3.2.1   $ETAS_0$, $ETAS_0^{b+}$

These two models are trained using the general ETAS method, as introduced in Sect. 3.1, on the ESHM20 catalog filtered as described in Sect. 2. Although the full training period includes data from between 1980 and 2015, the first 10 years are used as a "burn-in period" – these events are interpreted as potentially triggering events, but we do not consider them as possible aftershocks of previous earthquakes. Without this "buffer" time period, the events near the beginning of the selected time window would all be interpreted as background events, having no preceding seismic activity acting as their potential triggering events. Additionally, this same auxiliary period will be used when simulating catalogs for the purpose of retrospective consistency testing, since in the simulated catalogs starting in 1990, we need both background events and aftershocks of seismicity that occurred prior to 1990.

Having a unique set of parameters for all of Europe provides a harmonized model that describes the aftershock behavior in the region. For each event, the EM parameter inversion algorithm also yields the probability that it is a background or a triggered event. This allows us to capture the spatial variations in rates of background events despite the fact that the background parameter $\mu$ is treated as a constant during the parameter inversion. When simulating catalogs that are later used for producing retrospective and pseudo-prospective forecasts, we use this background rate information by drawing the locations of background events generated for the simulation period based on the probabilities that each observed event in the training catalog is a background event.

### 3.2.2   $ETAS_{bg}$, $ETAS_{bg}^{b+}$

We mentioned in Sect. 3.2.1 that the probabilistic branching structure inferred during the parameter inversion stage can be used to simulate new catalogs that will be consistent with the observed background event rates at different locations. However, a desired property of our model would be the ability to include the knowledge about variation in the background rate already during the parameter inversion in order to estimate the background probabilities of events more correctly and also to achieve consistency of the background seismicity rates with the ones in the long-term hazard model (Danciu et al., 2021). This means that the time-independent seismicity rates provided by the hazard model should match the time-dependent ones when computed over very long periods of time. For this reason, the model is adjusted to allow for the now space-varying background rate:

$$\ell(t, x, y) = \mu(x, y) + \sum_{i:t_i < t} g(m_i, t - t_i, x - x_i, y - y_i), \quad (4)$$

which similar to the ETAS formulation with an inhomogeneous background rate $\mu(x, y)$ suggested in Veen and Schoenberg (2008), where $\mu$ is modeled by subdividing the spatial observation window into $n$ cells with a constant-background rate $\mu_k$, $k = 1, \dots n$. Here, we allow for the variation between different locations to be fixed to the levels given as input to the ETAS parameter inversion. More precisely, during the expectation step in the calculation of the probability that one event triggered another one, the term $\mu$ representing the background contribution is replaced with

$$\mu(x, y) = \iota \cdot \mu_k, \tag{5}$$

where $\iota$ is a parameter learned in the inversion, estimated in every iteration as the total number of background events in the entire area, and normalized per day and square kilometer and $\mu_k$ is the long-term annual seismicity rate given as an input corresponding to the cell that contains the location $(x, y)$. The probability that event $i$ triggered event $j$ estimated in the $(n + 1)$th iteration is then given as

$$P^{(n+1)}(i \to j) = \frac{g(m_i, t_j - t_i, x_j - x_i, y_j - y_i|\theta^{(n)})}{\iota \cdot \mu_{k:j \in \text{cell } k} + \sum_{i:t_i < t_j} g}, \tag{6}$$
$$(m_i, t_j - t_i, x_j - x_i, y_j - y_i|\theta^{(n)})$$

where $\mu_{k:j \in \text{cell } k}$ is the long-term annual seismicity rate given as an input corresponding to the cell that contains the location $(x_j, y_j)$.

As $\iota$ is estimated in each iteration as the total estimated number of background events per day and square kilometer, it represents the overall background rate. Therefore, the information that needs to be taken from the input background level is not the absolute background rate in the corresponding spatial cell, normalized per time and area unit, since multiplying two such values would result in a quick convergence of this parameter to 0. Rather than that, we only take the relative relationship between these rates among different spatial cells by normalizing the values $\mu_k$ before inversion so that $\frac{1}{n}\sum_{k=1}^{n} \mu_k = 1$, where $n$ is the number of $0.1° \times 0.1°$ cells that cover the area of interest, in our case $n \approx 8 \times 10^5$. For each event in the catalog, we assign the corresponding background seismicity level within its respective bin, which is then used as $\mu_k$ during the inversion.

### 3.2.3 ETAS$_\alpha$, ETAS$_\alpha^{b+}$

In the literature, it has been observed that there is a tendency in ETAS models to underestimate the productivity of large events (Hainzl et al., 2013), possibly due to anisotropy of the aftershocks compared to the isotropic model (Helmstetter et al., 2005; Hainzl et al., 2008; Zhang et al., 2020), covariance between the productivity parameters (Sornette and Werner, 2005), or their underrepresentation in training data. Therefore, another proposed modification of the model is to allow the productivity term $\alpha = a - \rho \cdot \gamma$ to be fixed to a given constant. This term emerges from our ETAS formulation as

the exponent in the relationship between the magnitude of an event and its expected number of aftershocks.

While the productivity law describes an increase in the number of aftershocks with the magnitude of the main event, the GR law explains that there are relatively fewer high-compared to low-magnitude events. As described in Helmstetter (2003), the relationship between the two exponents of these exponential relationships, $\alpha$ and $\beta$, determines whether earthquake triggering is driven by low- or high-magnitude events and stipulates that $\alpha = \beta$ balances the influence of events of different magnitudes in earthquake triggering.

In the ETAS$_\alpha$ model variant, we apply this fixed $\alpha$ during the inversion based on the EM algorithm, which naturally affects all parameters. For this modification of the ETAS model, we set $\alpha = \beta$, as suggested in van der Elst et al. (2022) when $b < 1$, where $\beta$ is the GR parameter. When estimating the $b$ value with the $b$-positive method (van der Elst, 2021) for the ETAS$_\alpha^{b+}$ model variant, we obtain $b > 1$. Therefore, following the recommendation by van der Elst et al. (2022), in order to prevent the "exploding" behavior of aftershock triggering, we fix the productivity term to $\alpha = \ln(10)$ (equivalent to fixing $a = 1$ in the base-10 formulation).

### 3.2.4 ETAS$_{\text{USGS}}$

The prior models used by the USGS AftershockForecaster software are fitted separately for different tectonic regimes; hence, more than one set of parameters exists. The sequences in the European dataset originate from various tectonic regimes; therefore we use their "global average" set of parameters. These parameters are expressed in an ETAS formulation we call standard, as it is more common in the literature (Ogata, 1992; Omi et al., 2014; van der Elst et al., 2022), where the temporal decay is given as $(\Delta t + c)^{-p}$, as opposed to the formulation in Eq. (1) used here, where the temporal decay is described by the factor $(\Delta t + c)^{-(1+\omega)} \cdot e^{-\Delta t/\tau}$. The productivity law in the standard formulation (van der Elst et al., 2022) is expressed as $10^{-\alpha(m_i - m_c)}$, where $m_i$ is the magnitude of the triggering event, and in Eq. (1), it is given as $k_0 e^{a(m_i - m_c)}$ but is influenced by the spatial kernel term $(\Delta x^2 + de^{\gamma(m - m_c)})^{-(1+\rho)}$. As no spatial parameters are specified in the parameter set given in the AftershockForecaster software documentation, we use the spatial kernel inverted by ETAS$_0$. Additionally, we use $\mu$ and the background locations inferred by ETAS$_0$ because the USGS models are fitted only to aftershock sequences and do not account for background seismicity. Keeping in mind the different $m_c$ value, which in the USGS global average parameter set is 4.5, we translate the parameters into our formulation as in Mizrahi et al. (2024a). Note that our version of the model is a simplification of the actual model employed by the USGS AftershockForecaster software and is not meant to replicate it exactly; the aim of including the model in our study is solely

to assess the usefulness of locally calibrated parameters compared to globally calibrated ones.

## 3.3 Consistency testing

A basic set of tests that one can do to assess the consistency of the models with past data is defined by the Collaboratory for the Study of Earthquake Predictability (CSEP; Zechar et al., 2010; Savran et al., 2020, 2022a, b). Passing retrospective number, magnitude, space, and pseudo-likelihood tests would imply that a model forecasts the occurrence of a similar number of similar magnitude events at places where they were observed in the training data.

Based on the background event occurrence and aftershock-triggering laws inferred during the inversion of ETAS parameters, we simulate 10 000 synthetic catalogs for the training period (1980–2015), with the first 10 years serving as a burn-in period introduced in Sect. 3.2.1 and the actual simulated period starting in 1990. The simulation procedure was implemented in Mizrahi et al. (2023) following the detailed description in Mizrahi et al. (2021b) and accounts for higher-order aftershocks. First, the background events are simulated by drawing their count from a Poisson distribution with the mean corresponding to $\mu$, occurrence time from a uniform distribution, and magnitude from a GR distribution ($\beta$ estimated from the data). For models with no additional background information given as input, the locations of the background events are drawn from the locations of existing events (with a Gaussian-distributed uncertainty), weighted by their probabilities of being background events. For models with informed background introduced in Sect. 3.2.2, the same background input which is used during inversion is also used as the spatial distribution of simulated background events to ensure long-term consistency with ESHM20 assessments. Note that in both cases the total number of background events is distributed according to the ETAS inversion output, but their locations are drawn based on background probabilities inferred by ETAS inversion in the first case and uniformly within each grid cell ($0.1° \times 0.1°$, latitude–longitude) defined by ESHM20 in the latter case.

The first generation of aftershocks is simulated by generating aftershocks of all events in the "starting" generation – their number, location, timing, and magnitude are determined by the productivity law, spatial decay, temporal decay, and GR law, respectively. Further generations of aftershocks are simulated iteratively by simulating aftershocks of all events in all previous generations until the number of events in the new generation becomes 0. Here, the auxiliary burn-in period (see Sect. 3.2.1) from 1980 to 1990 of the observed catalog is used together with a set of simulated background events between 1990 and 2015 as a starting generation of events. For all models, the maximum magnitude during the simulation phase is set to $m_{max} = 10.0$, which, due to the binning value of $\Delta m = 0.2$, corresponds to $m_{max} = 10.1$. Magnitudes are simulated based on the GR law with the $b$ value estimated

with adjustment for rounded values and binned to 0.2 to be consistent and comparable to the observed (training) catalog.

The number test ($N$ test) consists of counting the number of events in each catalog to get an approximation of the distribution of the forecasted number of events, which is then checked against the observed number of events in the true (observed) catalog. The quantile score of the test is computed as the probability of observing the true number of events under the assumption that the number of events follows the distribution approximated by the simulations. This hypothesis is then rejected when the quantile score is below 0.05 or above 0.95 (extreme 10 % of the forecasted distribution).

Similarly, the magnitude test ($M$ test) and the space test ($S$ test) compare the number of observed and forecasted events taking into account their magnitudes and locations, respectively. In the magnitude test, the distribution of deviations of each simulation's magnitude distribution from a "theoretical" magnitude distribution described by the set of all events across all simulations is compared to the same deviation for the magnitude distribution in the observed catalog. This deviation is calculated as the sum of squared logarithmic residuals between the normalized observed magnitudes and the theoretical magnitudes' histogram. Both when estimating the $b$ value and here, because of differences in $m_c$ in space and time, we observe $m - m_c(x, y, t)$ instead of pure magnitudes, as these differences should follow an exponential distribution.

In both spatial and pseudo-likelihood (PL) tests, the property of interest in the simulated catalogs and the observed catalog is not their length (as in the $N$ test) nor a metric describing the deviation of a magnitude distribution from the theoretical one (as in $D^*$ statistic for the $M$ test) but rather the pseudo-likelihood computed as the sum of the approximate rate density over all spatial bins. The pseudo-likelihood test combines space–magnitude gridding to obtain an overall comparison of the consistency between forecasted and observed catalogs. Unlike the number test, these tests are defined as one-sided, meaning that the hypothesis that the true magnitude or spatial distribution follows the one in simulations is only rejected when the quantile score is above 0.9 in the $M$ test or below 0.1 in the $S$ test and PL test.

In addition to performing the $N$ test, to compare the observed and modeled overall count of events during the training period (1990–2015) in more detail, the same consistency check can be done for smaller subsets of this time interval. Here, we perform this check for the cumulative count of events in increasing time intervals, all starting at the beginning of the training period, by comparing the observed to simulated counts of events between 1990 and every year in the training period. As in the $N$ test, we consider the model to be consistent with the observation at any given point if the observed count of events at that point falls within the 90 % confidence interval, bounds of which are estimated by the 5th and 95th percentiles of the cumulative counts of events in the simulated catalogs.

Due to the varying completeness magnitude, each event is given a weight during the inversion of ETAS parameters correcting for the estimated number of unobserved events at the time and location of that event. The simulated catalogs contain events above $m_{\mathrm{ref}}$, the minimum $m_{\mathrm{c}}$ across the entire catalog, while the observed catalog only contains the events above the corresponding $m_{\mathrm{c}}(x, y, t)$. To make the synthetic catalogs comparable with the observed catalog, we cut off the synthetic catalogs to contain only events above the corresponding $m_{\mathrm{c}}(x, y, t)$ values.

While the same tests can be performed pseudo-prospectively, meaning with test data that the model was not trained on (in our case, that is the data after 2015), to check for consistency with the training data, we focus on performing the tests retrospectively. Apart from providing a sanity check and indicating potential shortcomings of a model, retrospective consistency testing enables evaluating its performance on long-term data, which is not available in the post-training time period (in our case 7 years versus the 35-year-long training period). These tests are performed for each model separately.

### 3.4 Pseudo-prospective testing

To compare the performance of the models in terms of their forecasting power, we set up a pseudo-prospective forecasting experiment. Each model is used to simulate 100 000 synthetic catalogs for consecutive 1 d testing windows in the 7-year-long testing period. The simulations are created similarly to the procedure described in Sect. 3.3, with the starting set of events consisting of the full training catalog and the portion of the testing catalog up to the time window for which the forecast is made. The aftershocks of all these events are then simulated based on the modeled aftershock behavior to create the first generation of aftershocks, and further generations are simulated iteratively until convergence. For each time window, the corresponding simulations are used to find a distribution of the number of events in each spatial bin.

Having estimated the forecasted distribution of the number of events in each spatial bin ($j = 1, \ldots, N$) for every time window (indexed with $i$), the forecast can now be compared to reality by checking the probability of the observed number of events in that space–time bin ($n_{i,j}$) given by the estimated distribution. This is done for each spatial bin and then summed over all spatial bins, resulting in the log-likelihood score of a model for a forecasting time horizon given as in Nandan et al. (2019) and Nandan et al. (2022):

$$\mathcal{LL}_{\mathrm{model}}^{i} = \sum_{j=1}^{N} \ln \left( P_{\mathrm{model}}^{i}(n_{i,j}) \right). \tag{7}$$

Note that when the estimated probability of $n_{i,j}$ events occurring in a spatial bin is 0, this log-likelihood score would not be finite. For this reason, after simulating the synthetic catalogs and observing the distribution of the number of events in each spatial cell, we slightly alter this distribution by taking a small probability ($\sim 10^{-7}$), called the "water level", and distribute it over the bins (up to a maximum bin $n_{\mathrm{max}}$) with a zero count, adjusting the event counts in all other bins to retain the property that the sum of probabilities of all event counts is 1. If this water level is too high, the originally simulated distribution will be overwritten by one that is closer to being uniform. Correct high-probability forecasts would thus receive a substantially lower log-likelihood score. On the other hand, if the water level is a very low value, we penalize its usage heavily. Although the score should reflect the fact that the model failed to forecast the observation, a penalty larger by many orders of magnitude than all other log-likelihood score differences would overrule the differences between models in all other observed bins.

The range of water level values that allow us to meaningfully distinguish models is chosen using the first 2 years of the testing dataset as an initial validation set. For this, we visually inspected the results obtained with different water levels with the first 2 years of data, using plots similar to those in Fig. S4. We then eliminated water level ranges which yielded undesirable behavior. For instance, for some water levels, the ETAS models score lower than the time-independent model both when events do and do not occur. Too high a water level leads to the subtraction of too large a number from the probability in non-empty event count bins, yielding much lower scores when water level is not used, while still scoring lower than the benchmark when water level is used, even though it is high, due to the uniform distribution of water level over remaining bins. Water levels being too low on the other hand and the fact that the water level is distributed uniformly over all empty bins result in an extremely low score when "unforecasted" event counts do occur. In our selection of the water level, we ensure that the bins that use the water level achieve a score still orders of magnitude lower than that of the benchmark (time-independent Poissonian) model, which assigns a non-zero probability everywhere, while not being overly penalizing when events do occur.

Within this experiment, the spatial bins are set to $0.1° \times 0.1°$ (latitude–longitude); the time window is 1 d; and events with magnitudes 4.6 and above are considered, which is the generally valid completeness magnitude in the testing part of the catalog. Since the experiment is pseudo-prospective, the new part of the catalog is available and 7 years of data since 2015 can be used for validation and testing, resulting in 2558 testing windows for which each model produces 100 000 synthetic catalogs.

As mentioned earlier, the baseline against which all model variants are tested is the Poisson background model, for which generating synthetic catalogs is not needed. Within each spatial cell, the number of events is considered to follow a Poisson distribution with a mean of $\lambda_j = \mu_{j,\mathrm{ESHM}}$, where $\mu_{j,\mathrm{ESHM}}$ is the daily seismicity rate in the $j$th spatial cell

given by ESHM20. The log likelihood in Eq. (7) becomes

$$
\begin{aligned}
\mathcal{LL}^i_{\text{ESHM20}} &= \sum_{j=1}^{N} \ln\left(P^i_{\text{ESHM20}}(n_{i,j})\right) \\
&= \sum_{j=1}^{N} \ln \frac{\lambda_j^{n_j} e^{-\lambda_j}}{n_j!}, \quad \text{for every time window } i. \quad (8)
\end{aligned}
$$

The metric used for comparison of the models is simply the difference between their log-likelihood scores, called the information gain (IG). For each time window, we have a value of IG of one model over another, and while the cumulative information gain is indicative of models' performance through time, we test whether one model outperforms another by testing whether the mean information gain (MIG) between that pair of models is significantly positive using a paired one-sided $t$ test.

## 4   Results and discussion

### 4.1   Model fit

As mentioned in Sect. 3, fitting an ETAS model to the data means finding a unique set of parameters describing the observed aftershock-triggering behavior. The set of inverted parameters for each of the described ETAS model variants is given in Table 1. The parameter $\mu$ describes the overall rate of background events, and it is estimated by counting the total number of background events and normalizing it per day and square kilometer. The count of background events is obtained by summing the probabilities $p_{\text{BG}}$ that are assigned to each event during the parameter inversion, weighted by the estimated ratio of unobserved and observed events $\zeta$ introduced in Sect. 3.1 to account for incompleteness.

As the parameters can be grouped into those describing temporal decay, spatial decay, and the productivity law, the curves of each can be plotted separately as in Fig. 2. These curves represent the modeled aftershock-triggering behavior and are compared to the observed aftershock-triggering behavior in the observed catalog, represented by dots. However, since the true triggering relationship between events in the observed catalog is unknown, for counting aftershocks triggered by an event of a certain magnitude at a given temporal and spatial distance, we rely on the probabilistic triggering structure inferred during the expectation step of the EM algorithm. Therefore, the observed aftershock-triggering behavior is, in fact, dependent on the inverted triggering parameters.

In Fig. 2a, the curves show the temporal decay in aftershock behavior described in ETAS as

$$
N(\Delta t) = \frac{\exp(-\Delta t / \tau)}{(\Delta t + c)^{(1+\omega)}}, \quad (9)
$$

and the dots represent the "observed" aftershock behavior by showing counts of pairs of events $(i, j)$, where $i$ triggered $j$ with probability $p_{ij}$ and $\Delta t = t_j - t_i$, computed as $\sum_j p_{ij} \cdot \zeta(j)$. In the top row, the different curves represent triggering laws that were inferred on different datasets: the European catalog used in the present study, Swiss seismicity (Mizrahi et al., 2024a), Californian seismicity (Mizrahi et al., 2021a), and parameters used by the USGS AftershockForecaster software (spatial kernel is taken from $\text{ETAS}_0$ as mentioned in Sect. 3.2.4); in the bottom row, the different curves represent the laws inferred by different ETAS variants. In a similar fashion, the spatial decay depicted in Fig. 2b shows the number of triggered aftershocks at distance $\Delta x$ described in the ETAS model as

$$
N(\Delta x) = ((\Delta x)^2 + d \exp(\gamma(m - m_{\text{c}})))^{-(1+\rho)}. \quad (10)
$$

As there is a dependency of the spatial decay on the magnitude, there is a curve describing this law for each magnitude bin. In Fig. 2b, we show $m = 4.0$. The line in Fig. 2c shows the dependency of the number of triggered events on the magnitude of the triggering event, described in ETAS formulation with the productivity law

$$
N(m) = k_0 \exp(a(m - m_{\text{c}})). \quad (11)
$$

### Discussion of the model fit

Comparing multiple models trained on the European dataset based on the ETAS parameters shown in Table 1, we consistently observe that the background term $\mu$ is higher in models that allow for background term variation during the inversion. This is in agreement with the idea that using an informed-background term $\mu$ during inversion allows models to recognize more events in active areas as background events, while they would be interpreted as triggered events (triggered by other events in the same active area) without the added background information (Nandan et al., 2021). That more events are interpreted as background events rather than aftershocks also manifests in the fact that informed-background model variants have lower overall productivity. This is seen in the branching ratio $\eta$, which reflects the average number of aftershocks per triggering event being lower for the background-informed models when compared to their constant-background counterparts. Furthermore, the lines in the bottom row of Fig. 2c are almost parallel in between informed and non-informed-background versions of the same model variants, but the line describing the informed-background variant is always below the line for the corresponding model without an informed background.

While the background-informed model variants have overall lower productivity than their equivalents without background information used during the inversion, another, more obvious difference in the productivity law is seen between the corresponding models with and without a fixed $\alpha$ term. Fixing this parameter to the GR $\beta$ value directly affects the

**Table 1.** Inverted ETAS parameters for each of the eight ETAS variants described in Sect. 3. Additional parameters include the $b$ value, productivity term $\alpha = a - \rho\gamma$, and the branching ratio $\eta$.

| Model | $\mathrm{ETAS}_0$ | $\mathrm{ETAS}_\alpha$ | $\mathrm{ETAS}_{bg}$ | $\mathrm{ETAS}_{bg,\alpha}$ | $\mathrm{ETAS}_0^{b+}$ | $\mathrm{ETAS}_\alpha^{b+}$ | $\mathrm{ETAS}_{bg}^{b+}$ | $\mathrm{ETAS}_{bg,\alpha}^{b+}$ | $\mathrm{ETAS}_{\mathrm{USGS}}$ |
|---|---|---|---|---|---|---|---|---|---|
| $\log_{10}\mu^*$ | $-7.94$ | $-8.08$ | $-7.24$ | $-7.22$ | $-7.75$ | $-7.91$ | $-7.05$ | $-7.04$ | $-7.94$ |
| $\log_{10}k_0$ | $-1.63$ | $-2.51$ | $-1.40$ | $-2.07$ | $-1.63$ | $-2.50$ | $-1.39$ | $-1.98$ | $-2.63$ |
| $a$ | $1.59$ | $3.11$ | $1.79$ | $3.27$ | $1.68$ | $3.17$ | $2.05$ | $3.39$ | $2.87$ |
| $\log_{10}c$ | $-2.65$ | $-3.01$ | $-2.37$ | $-2.43$ | $-2.58$ | $-2.90$ | $-2.27$ | $-2.32$ | $-2.57$ |
| $\omega$ | $-0.11$ | $-0.15$ | $-0.04$ | $-0.05$ | $-0.10$ | $-0.14$ | $-0.02$ | $-0.03$ | $-0.03$ |
| $\log_{10}\tau$ | $3.66$ | $3.9$ | $3.44$ | $3.78$ | $3.67$ | $3.91$ | $3.46$ | $3.80$ | $12.26$ |
| $\log_{10}d$ | $0.92$ | $0.54$ | $0.90$ | $0.69$ | $0.90$ | $0.53$ | $0.86$ | $0.70$ | $0.92$ |
| $\rho$ | $0.61$ | $0.55$ | $0.81$ | $0.82$ | $0.64$ | $0.57$ | $0.87$ | $0.89$ | $0.61$ |
| $\gamma$ | $0.92$ | $1.52$ | $0.88$ | $1.20$ | $0.94$ | $1.51$ | $0.96$ | $1.22$ | $0.92$ |
| $b$ | $0.99$ | $0.99$ | $0.99$ | $0.99$ | $1.23$ | $1.23$ | $1.23$ | $1.23$ | $1$ |
| $\alpha$ | $1.03$ | $2.28$ | $1.08$ | $2.28$ | $1.08$ | $2.30$ | $1.21$ | $2.30$ | $2.28$ |
| $\eta$ | $1.00$ | $4.46$ | $0.75$ | $3.03$ | $0.83$ | $0.97$ | $0.60$ | $0.65$ | $4.04$ |

* Spatially varying, showing the approximated average ($\iota$ in Eq. 5).

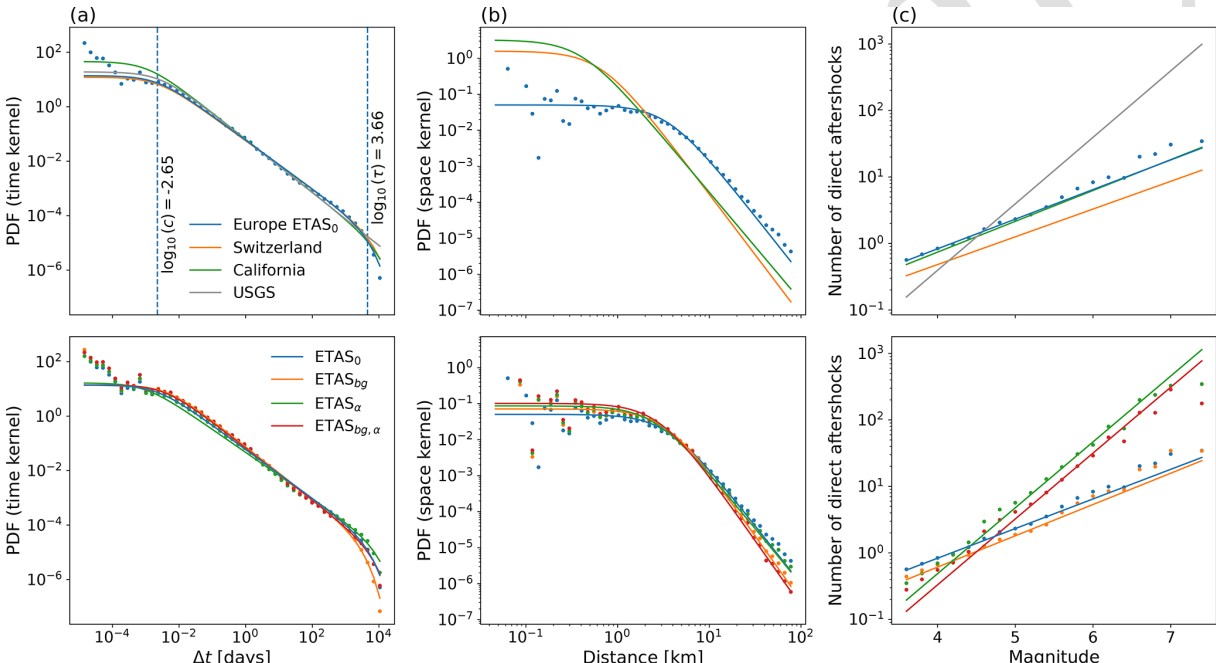

**Figure 2.** Plots of the model fit. In the first row, triggering laws inferred with the $\mathrm{ETAS}_0$ model are shown, including lines representing models for other areas for comparison. In the second row, modifications introduced in Sect. 3.2 are compared. **(a)** Temporal decay. **(b)** Spatial decay. Due to the dependency of the spatial decay law on the magnitude of the triggering event, there is a curve describing this law for every magnitude, here $m = 4.0$. **(c)** Productivity law. PDF: probability density function.

productivity law plot as the slope of the lines is exactly $\alpha$, resulting in a steeper slope which indicates relatively higher productivity assigned to high-magnitude triggering events compared to low-magnitude triggering events. Fixing $\alpha$ also drastically increases the branching ratio $\eta$. In non-informed-background model variants, this increase in productivity is counterbalanced with a lower background rate. Interestingly, the informed-background model variant with the standard $b$-

value estimate shows increased productivity and an increased background rate when $\alpha$ is fixed. Unlike for $\mathrm{ETAS}_\alpha$ and $\mathrm{ETAS}_{bg,\alpha}$, in models with a fixed $\alpha$ and $b$-positive estimate of the $b$ value, the branching ratio remains lower than 1 due to $\alpha$ being fixed to $\ln(10)$ instead of $\beta$ (van der Elst et al., 2022), since $b > 1$.

Furthermore, model variants relying on the $b$-positive estimate of the $b$ value consistently display a higher back-

ground seismicity level and lower overall productivity but a higher productivity term $\alpha$ than their counterparts using the traditional $b$-value estimator. All these differences must be due to what the model infers about unobserved events below $m_c(x, y, t)$ from the observed ones by applying the GR law. A higher $b$ value leads to a larger number of expected unobserved aftershocks of observed events. Thus, when $m_c(x, y, t) > m_{ref}$, the observed high-magnitude events' productivity is inflated, while lower-magnitude events' productivity is not (or less) inflated, explaining the larger $\alpha$ for $b$-positive variants. The higher background rate and lower branching ratio of $b$-positive variants suggest that the parts of the catalog that are less complete and hence more inflated when the $b$ value is high exhibit this behavior. To decide whether this is caused by differently behaving seismicity in certain areas or time periods or whether this indicates that seismicity behaves differently for higher-magnitude events compared to lower-magnitude ones, further research is required.

Apart from differences among models in interpreting events as background or triggered, a trend in some of the other parameters can also be observed. In the spatial distribution of aftershocks, models with fixed productivity tend to have higher $\gamma$ but lower $d$ values, interpreting more of the events at shorter distances as aftershocks of low-magnitude events and more of events further away as aftershocks of high-magnitude events. For the temporal distribution of aftershocks, the higher $\omega$ in background-informed models implies a slower decay of the number of aftershocks, whereas the fixed productivity variants have lower $\omega$ values, resulting in a faster decay. However, $\tau$ is larger for fixed productivity variants and lower for background-informed variants, meaning that the tapering of the distribution will occur later in the former case (after about $10^{3.66}$ d, which corresponds to approximately 12.5 years) and sooner in the latter ($10^{3.44}$ d or around 7.5 years).

When comparing the laws described by ETAS models' parameters, we observe differences not only among model variants introduced in our study but also among models calibrated on distinct datasets. While the time kernels appear quite similar across models for different regions, with the inclusion of multiple "European" models, a notable disparity arises in the spatial kernel when comparing the European models to those trained for Switzerland (Mizrahi et al., 2024a) or California (Mizrahi et al., 2021a), as can be seen in the top row of Fig. 2b. Specifically, we note a lower frequency of observed aftershocks at shorter distances from the triggering event, with a subsequent decay starting at slightly greater distances. Since we observe this difference when comparing to models calibrated on other datasets but not when comparing multiple European models, one possible explanation for this observation could be differences in the location determination of events, due to lower location precision and possible short-term incompleteness (close in time and space to triggering events) in some areas in this highly heterogeneous catalog. This idea is supported by the fact that the difference between spatial kernels diminishes with an increase in the magnitude of the triggering event.

Additionally, the comparison between the ETAS variants fitted to the European dataset and the adjusted ETAS parameters used by the USGS AftershockForecaster software reveals that the productivity law is more similar to those inferred by ETAS variants with fixed $\alpha$ values, which is expected given that $\alpha$ is fixed to 1 (log base 10, in our formulation this corresponds to $\alpha = \ln 10$) in all sets of parameters used by USGS models. Another difference is in the temporal kernel, which in the case of $ETAS_{USGS}$ does not have a tapered exponential form. This results in relatively more aftershocks forecasted for periods long after the triggering event; in Table 1, all models have a $\tau$ value between $10^3$ and $10^4$, whereas in $ETAS_{USGS}$, it is set to $10^{12}$, corresponding to a period of 5 billion years (effectively, this means there is no taper).

## 4.2 Results of consistency tests

To visualize the output of CSEP consistency tests, the PyCSEP implementation Savran et al. (2022a, b) provides the option of displaying the modeled behavior of the events as a histogram created based on the set of a large number of synthetic catalogs given as the output of a model (catalog-based forecast) and comparing it to the observed value in the observed catalog represented by a dashed vertical line.

In our case, the distribution of the number of events is approximated by counting the lengths of all simulated catalogs and the vertical line is the size of the catalog which was used in training for the inversion of ETAS parameters. Figure 3a shows the visualization of the $N$ test for $ETAS_0$, with the histogram describing the model's distribution of the number of events, whereas the vertical line represents the number of events in the observed catalog.

The quantile scores for all tests and all model variants except for $ETAS_\alpha$ and $ETAS_{\alpha,bg}$, the retrospective simulations of which do not converge, are shown in Fig. 3b and represent the position of the dashed line showing the observed property with regards to its forecasted value shown by the histogram (see Fig. 3a). Quantile scores between 0.05 and 0.95 indicate that the $N$ test has been passed, while values below 0.05 or above 0.95 indicate that a model has failed the test. In case of the $S$ test and PL test, the tests are one-sided; therefore, the models with quantile scores above 0.1 pass the tests. The $M$ test is also one-sided but defined so that the models with quantile scores below 0.9 pass the test.

In Fig. 4, forecasted and observed cumulative counts through time are compared for all ETAS models inverted in the European dataset that produced converging retrospective simulations. At points in time where the true (observed) event count is within the blue-shaded area, we consider the forecast to be consistent with the observation. The comparison between the observed count curve and the shaded area at the

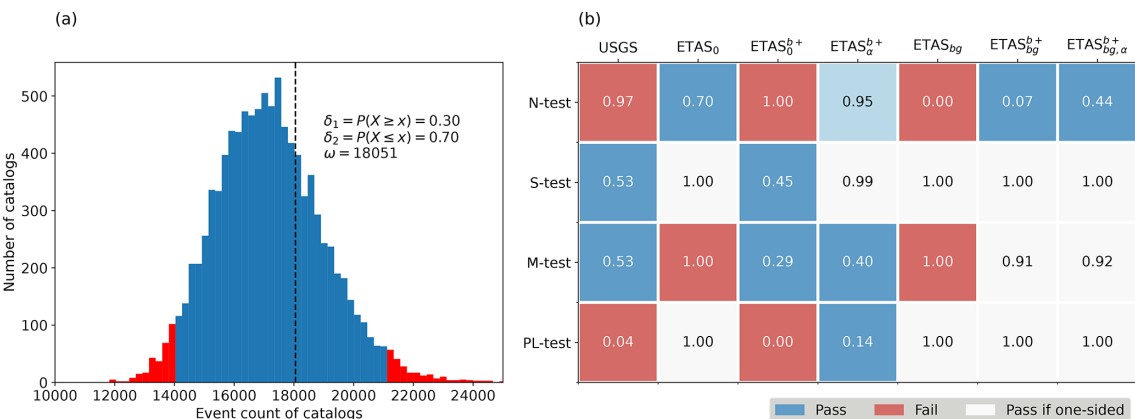

**Figure 3. (a)** The number test ($N$ test) for $ETAS_0$: histogram represents the modeled distribution of the number of events in the training period approximated by the counts of events in 10 000 simulated catalogs with completeness levels as in the training catalog. Dashed vertical line indicates the event count observed in the training catalog. **(b)** Table containing the quantile score of the models introduced in Sect. 3 for each of the consistency tests. Red color indicates failure (extreme 10 % quantiles): for two-sided tests ($N$ test), these are the lowest and highest 5 % quantile scores, and for one-sided tests, these are either scores above 0.9 (highest 10 %; $M$ test) or below 0.1 (lowest 10 %; $S$ test, PL test). Blue color indicates that the model passes the test; cells that are grey are ones where the model does not fail the test but due to the extreme quantile should be further investigated.

rightmost time point in every subplot corresponds with the $N$ test, the result of which is shown in Fig. 3.

Apart from the event count analysis in time shown in Fig. 4, the spatial distribution of the retrospective forecasts is visualized in Fig. 5 for models $ETAS_0$ and $ETAS_{bg}$ (upper row) and compared with the observed spatial distribution of the events in the training catalog during the same period (lower row). The color of each spatial cell in the maps corresponds to the mean number of events in that cell over 100 000 simulations. The number of events is counted during the full 25-year-long training period, and only events above $m_c(x, y, t)$ are considered.

## Discussion of the consistency test results

Due to the large branching ratio of $\eta > 1$ in ETAS variants with the productivity term $\alpha$ fixed to $\beta$, the number of events quickly explodes when simulating over a long-term period, as, on average, every event in the synthetic catalog will produce more than one aftershock in each generation of the simulation process. Therefore, the procedure does not converge and we consider the models $ETAS_\alpha$ and $ETAS_{bg,\alpha}$ to be failing the retrospective consistency tests. This suggests that our approach of fixing the productivity parameter $\alpha$ to $\beta$ during the inversion process is not suited to produce models that are consistent with reality in the long term, unless $b > 1$ ($\beta > \ln(10)$), in which case we fix $\alpha$ to $\ln(10) < \beta$, keeping the branching ratio below 1, even though $\alpha$ is fixed to a higher value than the one inverted without a constraint. Still, with the $ETAS_\alpha^{b+}$ and $ETAS_{bg,\alpha}^{b+}$ models, while the median number of events simulated is similar to that with $ETAS_\alpha$ and $ETAS_{bg,\alpha}$, respectively, some simulations contain more explosive sequences, resulting in a higher uncer-

tainty in the modeled number of events (Fig. 4, second and third columns). To avoid an underestimation of the productivity of large events without overestimating the overall productivity, differently parameterized productivity laws could be considered in the future. For instance, the logarithm of the aftershock productivity might be better described as increasing quadratically rather than linearly with the magnitude of the triggering event.

Furthermore, a branching ratio larger than 1 might be present during an ongoing sequence but is not sustainable over a longer term. Thus, considering sequence-specific parameters or distinct productivity for mainshocks and non-mainshocks, as done in the ETAS model employed by the USGS (van der Elst et al., 2022), would be a promising aspect to explore in the future. In $ETAS_{USGS}$, a high branching ratio is also observed that could result in such exploding behavior during synthetic catalogs' simulation. However, due to the untapered temporal kernel that forecasts a relatively higher, when compared to the tapered ones, number of aftershocks in periods even after $\tau \approx 10^3$ d, many of the simulated aftershocks will be assigned a timing outside of our period of interest, effectively resulting in a much smaller branching ratio. This effect is so significant that these simulations not only converge but also significantly underestimate the number of events. Mancini and Marzocchi (2023) also successfully fit an ETAS model with the productivity term fixed to $\beta$ but with no taper in the temporal kernel, resulting in a lower effective branching ratio than the one present in $ETAS_\alpha$ and $ETAS_{bg,\alpha}$.

In synthetic catalogs simulated based on the $ETAS_0$ model, around 30 % of the simulations have a higher event count than the observed value, and therefore the $N$ test is

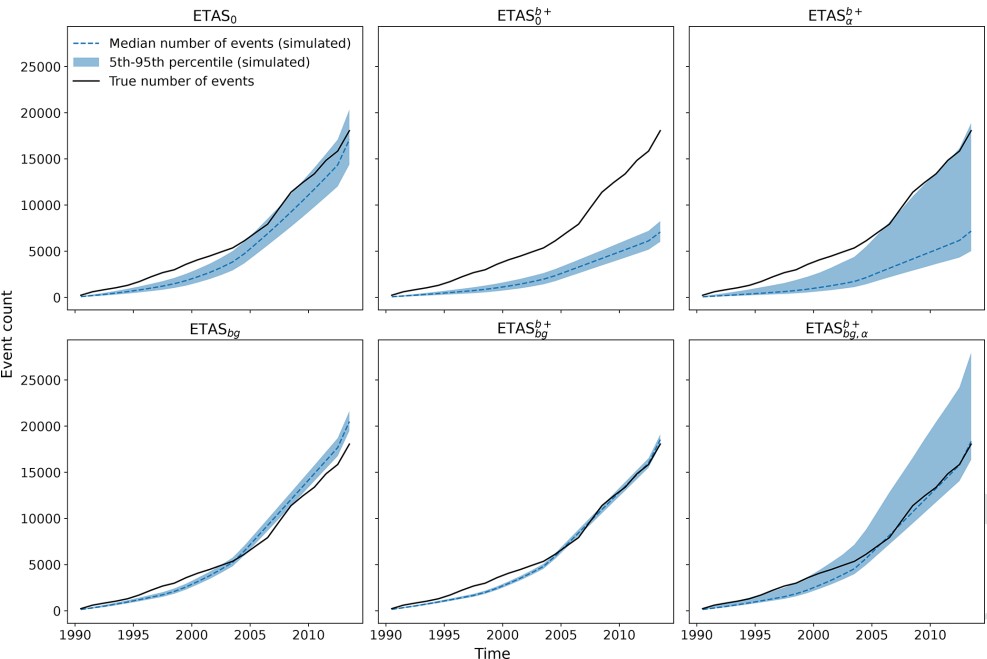

**Figure 4.** Cumulative count of events (median with 90 % confidence interval) simulated by each European model, compared to the observation. $ETAS_\alpha$ and $ETAS_{bg,\alpha}$, which fail to produce converging retrospective simulations, are excluded. $ETAS_{USGS}$ is shown in Fig. S3.

passed. However, the distance between the true observed magnitude distribution from the theoretical distribution estimated jointly from all the simulations is significantly higher than the distance between the observed magnitude distribution in each simulation and the theoretical distribution. The disagreement between the modeled and true magnitude distribution could be the result of either the true distribution not following a GR law with a single $b$ value but rather a mixed distribution with $b$ values varying in space and/or time or a bias caused by incompleteness potentially still present in the catalog. The same inconsistency in magnitude distributions can be observed in $ETAS_{bg}$, which applies the same $b$ value of around 0.99 but fails the number test by significantly overestimating the event count mostly due to the relatively low uncertainty.

The magnitude distribution seems better in the ETAS variants that rely on the $b$-positive method for $b$-value estimation, which all pass the $M$ test. This is consistent with a visual estimation that the $b$-positive ($b = 1.23$ in Fig. 1d) estimate fits the data better. However, due to a significant underestimation of the event count, $ETAS_0^{b+}$ fails the $N$ test, and $ETAS_\alpha^{b+}$ is on the significance threshold between passing and failing. On the other hand, $ETAS_{bg}^{b+}$ and $ETAS_{bg,\alpha}^{b+}$ pass all the consistency tests but with a much higher quantile in the $M$ test, implying a greater distance between the observed and forecasted magnitude distributions while using the same $b$-value estimate as $ETAS_0^{b+}$ and $ETAS_\alpha^{b+}$. This is most likely due to the dependence of the test on the event count in simulated

catalogs which is, as mentioned earlier, underestimated by $ETAS_0^{b+}$ and $ETAS_\alpha^{b+}$.

This correlation between the $N$ and $M$ test is being analyzed in more detail and avoided by modifying the $M$ test in Serafini et al. (2024); it is here further highlighted by the fact that $ETAS_{USGS}$, in contrast to $ETAS_0$ and $ETAS_{bg}$, passes the $M$ test, despite all models using very similar $b$ values of 1 and 0.99. A similar correlation is also present in the $S$ test, as evidenced by the fact that $ETAS_{USGS}$ and $ETAS_0$, which use an identical spatial distribution of background events and aftershocks, achieve significantly different scores. Except for models that significantly underestimate the event count ($ETAS_{USGS}$ and $ETAS_0^{b+}$) and therefore do not have reliable $S$-test scores, the quantile of the observed likelihood computed in spatial consistency tests is in the upper tail of the distribution of likelihoods of the synthetic catalogs. This indicates that the forecasts describe the observed data "suspiciously well", which does not necessarily imply failure of the test but suggests that the model requires further testing (Schorlemmer et al., 2007) due to being too smooth; i.e., the events occur too close to likelihood peaks without the expected scatter. Similar behavior is observed in pseudo-likelihood consistency tests as well. These findings suggest that the results of the $M$, $S$, and PL tests should be interpreted with caution and models should not be hastily rejected or accepted based solely on their performance in these specific tests when their $N$ test is failed.

Further insight into the modeled and observed event count consistency is provided by their cumulative comparison

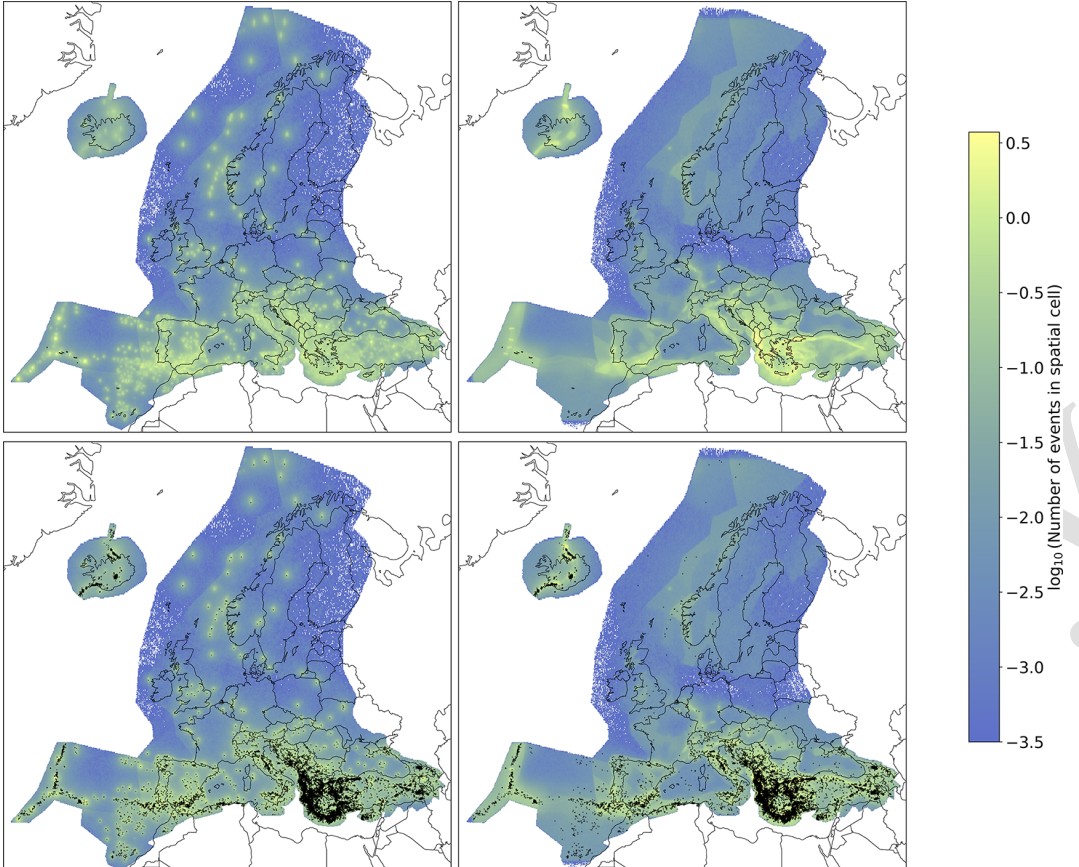

**Figure 5.** Comparison of the spatial distribution of events retrospectively forecasted by $ETAS_0$ (left) and $ETAS_{bg}$ (right) for the training period (1980–2015); color indicates the number of events in every spatial bin above the corresponding $m_c(x, y, t)$ for the duration of the entire 25-year period. In the bottom row, observed events are added to both maps. Note that events above $m_c(x, y, t)$ are shown for both the forecasts and the observations. White areas indicate no simulated events, with most areas outside the region of interest but also in some spatial bins within the region with zero events in 10 000 simulations.

through time shown in Fig. 4. In general, event counts begin to be more consistent starting with the year 2005, which coincides with the latest change in completeness magnitude in some of the regions (see Fig. S1). In more recent periods, when the differences in completeness levels between regions in the catalog are smaller, all models display consistency corresponding to their final $N$-test result: $ETAS_0$, $ETAS_{bg}^{b+}$, and $ETAS_{bg,\alpha}^{b+}$ forecast the event count as being consistent with the observation; $ETAS_0^{b+}$ heavily underestimates the event count; and $ETAS_\alpha^{b+}$ is on the significance threshold of underestimating it, with the median forecasted count being as low as for $ETAS_0^{b+}$. $ETAS_{bg}$ is the only model overestimating the event count, and it does so in every time period starting from 2005 and demonstrates a lower uncertainty range.

Overall, we can conclude that ETAS models with fixed $\alpha$ values not using the $b$-positive method to estimate the $b$ value clearly fail the long-term consistency tests. Among the remaining models, all models but $ETAS_0^{b+}$ and $ETAS_{bg}$ pass the $N$ test, which is often considered the most crucial of

consistency tests. $ETAS_{bg}$ and $ETAS_0$, which use the same $b$ value, fail the magnitude test, while the models which use the $b$-positive estimator pass it. The $S$-test scores do not provide a conclusive comparison between models, but a visual inspection of retrospective forecasts they produce shown in Fig. 5 suggests that background-informed models are less prone to overfitting the spatial distribution to existing events in the training catalog. The spatial distribution of $ETAS_0$ replicates the existing catalog because background events are simulated at a higher rate in areas where a higher background probability was inverted during the fitting of ETAS parameters. Thus, we can consider only variants applying the $b$-positive estimate and using ESHM20 background seismicity levels as adequate choices for a first harmonized ETAS model for Europe. The other consistency tests provide additional information about the potential limitations of these models, which shall be addressed in future efforts to improve the models, most important of which is the potentially oversimplified magnitude distribution applying a single $b$ value.

### 4.3 Results of pseudo-prospective tests

To compare the pseudo-prospective performance of 1 d forecasts issued by the models, we visualize their cumulative information gain in Fig. 6a. As introduced in Sect. 3.4, the log-likelihood score of a model is the logarithm of likelihoods summed over space, and it is therefore always negative. The information gain is the difference between the log-likelihood scores of two models and is positive when the first model assigned a higher probability of the actual occurrence than the second model. The reference log-likelihood score to which we compare others in Fig. 6a is the one of the Poissonian time-independent base model. To determine whether one of the models is significantly outperforming another one, we apply the paired one-sided $t$ test to the information gain values of the individual forecasting periods. In this way, we decide for each model pair whether the mean information gain (MIG) between the two models over all testing periods can be considered significantly positive (Fig. 6d, color indicates the MIG and a dot indicates the significance in the level of outperforming, similarly to the way pairwise information gain is shown in Iturrieta et al., 2024). The information gain is also shown when computed over the entire region of interest as just one spatial bin in Fig. 6b, with the significance matrix shown in (e).

The most prominent observation is the positive mean information gain of all ETAS models when compared to the time-independent Poissonian model with an ESHM20-informed background and $p$ values below 0.05 showing that for all models except $ETAS_{bg,\alpha}$ and $ETAS_{USGS}$ this positive score is significant. The three best performing models are $ETAS_0$, $ETAS_{bg}^{b+}$, and $ETAS_0^{b+}$, which all significantly outperform all variants with a fixed productivity term and $ETAS_{bg}$, with this result being almost always significant or near the critical $p$ value according to the $t$ test. Another major result is that all ETAS-based models significantly outperform the time-independent model when no spatial binning is used, with variants with a fixed productivity term $\alpha$ achieving the best score.

To analyze the performance of the models more thoroughly, we observe the spatial component of their log-likelihood scores by visualizing the total information gain over time for each spatial cell separately. The maps in Fig. 7a–b show the information gain between $ETAS_0$ and the time-independent model and between $ETAS_{bg,\alpha}^{b+}$ and $ETAS_0$, respectively. The spatial cells are joined into larger ones for better readability of the map. The more active areas show more pronounced total IG values, while no distinct spatial trends can be observed.

### Discussion of the pseudo-prospective test results

That ETAS models outperform the time-independent model in 1 d forecasting experiments is a result that highlights the core strength of ETAS models of modeling the short-term clustering behavior of earthquakes. These models show a substantial improvement in predictive skill over the time-independent model, and this predictive skill shows up during periods of clustering (as can be seen when comparing Fig. 6a and c). The highest jump in information gain is observed in late 2018, coinciding with the occurrence of an earthquake with a magnitude of 6.8 in the Ionian Sea late at night, followed by 14 events on the next day and over 30 within the week. In contrast, a larger event with a magnitude of 7.0 in the Aegean Sea in late 2020 is not coupled with such a jump, possibly as it was followed by only 10 observed events during the first week, 4 of which occurred on the day of the main event, which happened in the middle of the day, and do not "participate" in evaluating the models, as forecasts are produced in a 1 d window, with a cutoff at midnight.

$ETAS_0$ and $ETAS_0^{b+}$ have the highest MIG values, followed by $ETAS_{bg}$ and the three remaining $b+$ variants, while the lowest scores by time-dependent models fitted on the European data are achieved by $ETAS_\alpha$ and $ETAS_{bg,\alpha}$. The two variants also failed the retrospective tests due to explosive behavior, but in the short term the score is relatively worse only in the spatially sensitive testing, indicating that the anisotropy is the main issue here, as discussed below. Due to observations made in the analysis of retrospective tests, preference should be given to models applying the ESHM20-informed-background rate map and models using the $b$-positive estimate. Both $ETAS_{bg}^{b+}$ and $ETAS_{bg,\alpha}^{b+}$ satisfy these constraints and also score well in the pseudo-prospective experiment, suggesting that they can be considered adequate choices for short-term earthquake forecasting. This is further supported by the result that no spatial trend in model performance can be identified, and thus the models do not seem to be overfitting certain particular subregions. $ETAS_{bg,\alpha}^{b+}$ also has the highest MIG score in the pseudo-prospective experiment with no spatial binning, making it a potential winner model, provided its spatial component is further analyzed and improved, in accordance with conclusions following below.

As mentioned in Sect. 3.4, in order to compute a log-likelihood score of a model for a day and spatial bin where none of its simulations placed events, a water level probability is distributed evenly over such bins. The information gain plot corresponding to the one shown in Fig. 6a for different water levels is given in Fig. S4. Our sensitivity analysis shows that the selection of this parameter influences the significance of the performance difference between the time-dependent models and the time-independent benchmark (which never uses the water level), but the order between ETAS variants is barely affected. While this dependency on a subjective choice is an undesired effect, the ability of a model to outperform the time-independent model for a range of water levels is not an artifact of the parameter because it was ensured that the information gain between any ETAS variant and the time-independent model is negative whenever the water level is used.

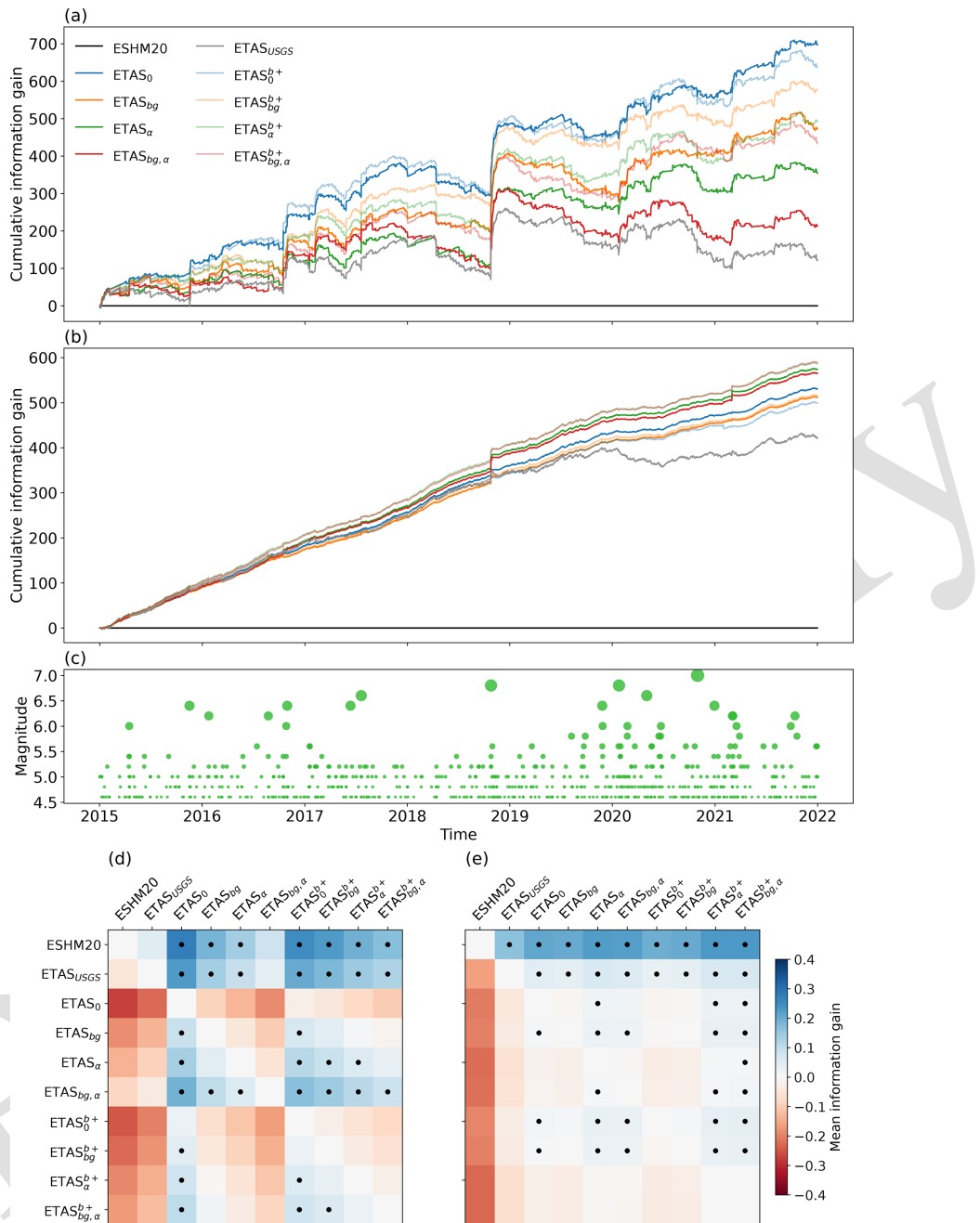

**Figure 6. (a–b)** Cumulative information gain through time, for consecutive non-overlapping 1 d windows over 7 years in the testing catalog. All models are compared to the Poissonian time-independent model, which acts as the null model. In **(a)**, spatial binning of $0.1° \times 0.1°$ (latitude–longitude) is applied. **(b)** No binning. **(c)** Seismicity in the testing catalog, with magnitudes through time. **(d–e)** Matrix of mean information gain of each model compared to all other models. Information gain in position $(i, j)$ compares the score of the model in column $j$ to that of the model in row $i$. Significant level of outperforming determined by paired one-sided $t$ tests for each pair of models is indicated by a dot.

Furthermore, in Fig. S5 we show the same information as in Fig. S4 but only apply the water level when the probability assigned by a model to the observed number of events is 0. While this is infeasible in a truly prospective setting as it is unknown whether or not the water level will be needed, it confirms that, for higher water levels, models lose according to score in number bins (most often 0) where simulations place a non-zero probability, but it is reduced to take the water level to distribute over other bins. However, the trend that still persists whether or not we only apply the water level

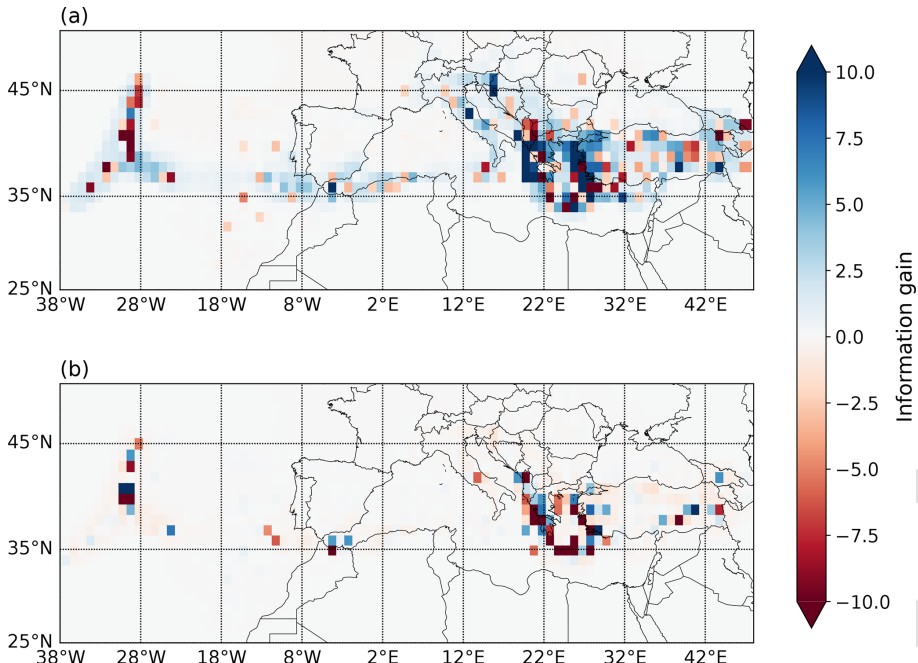

**Figure 7.** Total information gain per $1° \times 1°$ (latitude–longitude) spatial cell. Models compared: **(a)** $ETAS_0$ and time-independent Poissonian model and **(b)** $ETAS_{bg,\alpha}^{b+}$ and $ETAS_0$.

when necessary is that if the penalty is too high for none of the simulations producing the observed number of events, the time-dependent models overall lose according to score compared to the time-dependent benchmark.

To replace the water level with an actual model output, one should perform a larger number of simulations, posing a significant computational challenge. Other possibilities could be investigated in the future, with a promising strategy of replacing simulations recently proposed by Mizrahi and Jozinović (2024). Another way to avoid using the water level is to observe spatial bins large enough to contain a meaningful forecast even in the 1 d window we are observing at the expense of evaluating the spatial distribution at a high resolution. An extreme example is shown in Fig. 6b, where the information gain is computed solely based on the event count, treating all of Europe as a single spatial bin. Since there is no spatial binning, the water level is seldom used, and the information gain to the time-independent benchmark is more significantly positive than before.

However, in this case, the order of ETAS models is highly affected, with variants applying a higher $\alpha$ achieving the best scores, suggesting that they are forecasting the number of events better but placing them wrongly and thus being outperformed in the space-sensitive testing. Since the spatial distribution of background events does not differ between the two model groups (inverted $\alpha$ versus fixed higher $\alpha$), the performance difference must arise from the location of aftershocks. The use of isotropic kernels for the spatial distribution of aftershocks may lead variants with fixed $\alpha$ values to

place a higher number of aftershocks in circles with larger radius, in most of which nothing is observed. In the future, more complex models should be explored that place aftershocks in a more elliptical shape instead of a circular one or along known fault planes.

## 5 Conclusions

In this paper, we calibrated and evaluated multiple variants of ETAS models in a highly heterogeneous dataset which summarizes the recorded seismicity in Europe over the 35-year-long period between 1980 and 2015. The main result of 7 years of pseudo-prospective 1 d forecasting experiments is that ETAS-based models provide a significant information gain when compared to the time-independent benchmark model that underlies the ESHM20 hazard model. Additionally, the best-performing ETAS-based models inverted in the European dataset outperform the time-dependent ETAS model using globally calibrated parameters. Besides a basic ETAS variant, we propose several modifications that, during model calibration, allow for the use of additional spatial information or fixing the productivity term of the model formulation. We found that fixing the productivity term to a higher value, which is suggested in the literature to overcome the underestimation of the productivity of high-magnitude events, can result in a highly overestimated branching behavior of events when the $b$ value is relatively low. However, for short-term forecasts compared solely regarding the event count, these models achieve the best performance. Therefore,

in future studies, other techniques to address the estimation of the aftershock productivity of large events should be further explored, such as accounting for known relationships between tectonic setting and aftershock productivity (Dascher-Cousineau et al., 2020; Page et al., 2016; Davis and Frohlich, 1991; Marsan and Helmstetter, 2017), fitting and applying sequence-specific aftershock productivity parameters (as in van der Elst et al., 2022), or combining with more precise aftershock spatial distribution modeling (Field et al., 2017; Reverso et al., 2018). For ETAS variants that leverage information about the spatially varying background rate already during the inversion, we found that the inferred parameters differ from those inferred using the basic approach, resulting in more events being interpreted as background events and fewer as aftershocks. The background-informed ETAS variants do not outperform their counterparts with non-informed-background rates in pseudo-prospective testing, but in retrospective testing they demonstrate better behavior regarding visual inspection of the spatial distribution of events and achieve the best scores if combined with the $b$-positive estimate of the $b$ value. The background component contains additional information about long-term seismicity patterns, seismotectonic properties of the area, and seismicity in areas not represented in either the training or the testing parts of the catalog. For this reason, what could be the main strength of this method is hidden by the very limited size and time period of available testing data.

Retrospective long-term consistency tests provide an additional characterization of the strengths and weaknesses of each model variant, highlighting that some of the proposed model variants adequately capture the number of events over longer time periods. The magnitude consistency tests conducted for models using both the traditional maximum likelihood estimator of the $b$ value and the $b$-positive estimator favor the latter approach but highlight potential areas of improvement of the proposed models. The simplification of using a single $b$ value to describe the magnitude distribution of all of Europe, as well as the assessment of the space–time variation in the catalog completeness provided through ESHM20, may need to be revisited. Aside from the potential improvements of the issued forecasts by revisiting the completeness assessment, $b$-value variations, and strategies for sequence-specific model updating, our proposed models could be improved by adding further complexity, such as considering an anisotropic spatial kernel of aftershock behavior and utilizing information such as earthquake focal mechanisms or finite fault rupture models (Böse et al., 2023).

In the process of real-time dissemination of earthquake forecasts, developing the underlying model is only the first step. The forecasts produced as the output of the introduced models are yet to be tested in a truly prospective manner. Recent CSEP efforts to establish a standardized open experiment format and the corresponding software support for performing such tests have resulted in the formation of the floating experiments (Iturrieta et al., 2023), providing a suitable environment for future evaluation of the properties of proposed models.

Another challenge in delivering earthquake forecasts operationally is the fashion of doing so: visualization and communication of models' outputs are a topic of ongoing discussion among seismologists and communication experts (Field et al., 2016; Becker et al., 2018; Savadori et al., 2022; Schneider et al., 2023). Both the layout and content of the final products depend on the use case in terms of the areas for which they are developed and the end users they serve, ranging from the wider public to civil protection services to insurance companies. The main authority to communicate earthquake forecasts and act on them remains local agencies and experts with knowledge specific to their area of interest. The role of the pan-European models presented here is to provide a harmonized global alternative less limited by administrative borders and information in areas where it would otherwise not be available.

*Code and data availability.* The training catalog with completeness assessments per tectozone and rate maps used as input here were produced by ESHM20 Danciu et al. (2021) and are accessible at https://doi.org/10.12686/a15. The continuation of the catalog used for testing is available at https://doi.org/10.5880/GFZ.EMEC.2021.001 (Lammers et al., 2023). The ETAS inversion and simulation code used to train the models and generate the forecasts was developed for Mizrahi et al. (2021b) and is available via the Zenodo repository at https://doi.org/10.5281/zenodo.7584575 (Mizrahi et al., 2023).

*Supplement.* The supplement related to this article is available online at [the link will be implemented upon publication].

*Author contributions.* Conceptualization and investigation: MH, LM, SW. Formal analysis, methodology, and visualization: MH, LM. Funding acquisition, project administration, and resources: SW. Software: LM, MH. Supervision: LM, SW. Writing – original draft preparation: MH, LM. Writing – editing and review: all authors.

*Competing interests.* The contact author has declared that none of the authors has any competing interests.

ther geographical representation in this paper. While Copernicus Publications makes every effort to include appropriate place names, the final responsibility lies with the authors.

*Special issue statement.* This article is part of the special issue "Harmonized seismic hazard and risk assessment for Europe". It is not associated with a conference.

*Acknowledgements.* The authors wish to thank Shyam Nandan, Laurentiu Danciu, Aron Mirwald, and Nicholas van der Elst for their contributions, ideas, and feedback, as well as Maximilian Werner and an anonymous reviewer and for their constructive approach and many valuable suggestions to improve the study.

*Financial support.* This research has been supported by the EU Horizon 2020 program (grant no. 821115) and the Horizon Europe Research Infrastructures (grant no. 101058129).

*Review statement.* This paper was edited by Laurentiu Danciu and reviewed by Maximilian Werner and one anonymous referee.

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
