# Peer review of "Towards a Harmonized Operational Earthquake Forecasting Model for Europe"

_EGUsphere, 2023_

## Author Response (AR1)

**Towards a Harmonized Operational Earthquake Forecasting Model for Europe: Author's response**

Marta Han[1], Leila Mizrahi[1], and Stefan Wiemer[1]

[1]Swiss Seismological Service (SED), ETH Zurich

**Changes in the manuscript**

As a result of the reviewers' comments, we made a number of large and small modifications to our manuscript and the study as a whole. We thank the reviewers for their meticulous analysis and insights, which we believe contributed highly to the content of the paper. Here, we list the most important changes in the manuscript. Minor changes were done as outlined in the initial response to the reviewers, attached below for reference.

- Addressing a point made by reviewers, we identified an error in the definition of two of our proposed models, $\text{ETAS}_{\text{bg},\alpha}$ and $\text{ETAS}_{\text{bg},\alpha}{}^{b+}$, which fix the productivity $\alpha$, not to $\beta$, but in accordance with the recommendations by van der Elst et al. (2022) to $\ln 10$. This had an impact on the models' parameters, and as a consequence, all outputs of these models. Before, due to the non-converging simulation procedure, they were not producing retrospective synthetic catalogs, but the new models are now achieving good results in retrospective tests, and different scores in pseudo-prospective tests. These changes are made in Sections 4.2 and 4.3.

- Addressing the point made in RC2 about the study conducted in Serafini et al. (2024) and running the newly defined tests as a part of that study, we identified that the magnitude tests should be performed on differences between simulated magnitudes and corresponding $m_c(x, y, t)$, not pure simulated magnitudes. This modified the results of M-tests, and while the main conclusion remains that a single $b$-value is probably not good enough to describe what is likely a mixed exponential distribution with $b$-values changing in time and space, the new results clearly favour the $b$-positive estimate.

- We performed a sensitivity analysis to identify the effect of the selected water level on the outcome of the pseudo-prospective experiment, which

revealed an unwanted dependence on this parameter, shown now in a figure in supplementary material and discussed further in Section 4.3.

- In Section 4.2, due to somewhat inconclusive spatial consistency test result, we add a visualisation of the spatial distribution of simulated events, which looks significantly different for models based on using spatially-varying background rate input. Per request of both reviewers, also in this section we add the comparison of retrospective cumulative count of events for all models.

- Discussions in both sections 4.2 and 4.3 are modified and have somewhat different conclusions than before, favouring other models now and identifying the fact that some are better at forecasting the event count, but would benefit from a better description of the spatial distribution of aftershocks.

- Other changes in the text are made according to the reviewers' helpful suggestions, mostly they are less substantial than the major ones listed above. In the previously submitted response (blue) attached below, we address them point-by-point, and point to the previously uncertain changes to the manuscript (orange).

**Reply on RC1**

*This paper describes the first attempt to build an operational earthquake forecasting system for Europe.*

*My overall opinion on the paper is positive, but I think that the authors should address some points to make the paper more convincing and reproducible.*

*Below I list my main comments that should be addressed in a revised version.*

We would like to express our gratitude to the reviewer for dedicating their time and effort to constructively review the manuscript, and for the positive overall conclusion about our work. We recognize the need to take the proposed steps for improving the study.

*- The description of the quality of the homogeneous earthquake catalog is missing. The authors refer to the paper Danciu et al. (2021), but it would be good to show something about the homogeneity in terms of magnitude. Each agency uses different magnitudes and it could be cumbersome and really challenging to homogenize them. But the homogenization of the magnitudes is essential for the goal of this paper.*

*So, I suggest adding some more quantitative information about the catalogs, including the kind of magnitude adopted and how different magnitudes have been homogenized.*

We address this point in the revised version of the manuscript in Sect. 2 (Data). While homogenization of magnitudes and gathering data in general is not the contribution of this paper, but Danciu et al. (2021), we agree it is a challenging task and deserves more detailed description.

*- The authors use a binned magnitude of deltaM=0.2. It is not clear to me if this is an average that accounts for uncertainty in very old and very new earthquakes. In this case, the use of a mean value could not be appropriate.*

*In other words, what is the rationale of this choice (deltaM=0.2)? Moreover, the use of deltaM=0.2 has important consequences in terms of the b-value. Some papers show that the use of binned magnitudes introduce a bias in the b-value calculation. This could be important in simulating data for forecast and testing. Reading the paper, I cannot understand if the same binning has been maintained also for the newer earthquake catalog used for prospective testing. If not, the b-value calculated using deltaM=0.2 could not be appropriate for simulating data that will be binned with a different deltaM.*

*This point has to be analyzed in detail in a revised version.*

We address this point in the revised version of the manuscript in two parts: the choice of the binning in Sect. 2 (Data), as it stems from the catalog itself; and emphasizing the $b$-value calculation method adjusted for binning outlined in Tinti and Mulargia (1987) we only briefly mentioned in Sect. 3 (Methods) of the original manuscript. We add description of the $b$-value estimation with the

adjustment for binning already in Sect. 2 when referring to Figure 1d and also expand the description of applying the binning during catalog simulation phase in Sect. 3 (Methods).

*- The authors use very often the term "aftershock". I know that this is contained also in the name of the model (ETAS), but this could be very misleading, in particular for people working on seismic hazard that have a different definition of aftershocks (e.g., aftershocks can never be larger than the mainshock). I suggest replacing the term "aftershock" with the term "triggered earthquake" that is more appropriate for the ETAS model, which assumes that earthquakes can be divided only as background and triggered, not as fore-main-aftershocks.*

We correct the mentions of term aftershock in the manuscript or clarify the usage of the term to avoid the confusion.

*- In the introduction the authors write "There is not a unique agreed-upon best way to provide OEF...". It is not clear to me if the authors are talking about communication or about scientific output. For instance, Jordan et al. (2011) made the case in which OEF should be provided continuously, whereas some agencies provide this information only in some circumstances. Are the authors referring to that? Or to the challenging way in which probabilities can be communicated? In any case I would suggest being clearer on this point.*

We extend this statement in Introduction to address this point in the revised version of the manuscript.

*- As regards the "model fit", I am wondering why the authors do not provide the usual residual plot that shows visually if the model explains well the data. The authors use different plots, that could be ok and add more information, but they look like less informative than the residual plot (at least, this is my first impression).*

The main idea behind showing the fit was to also visualise the three aftershock behavior laws (spatial and temporal decay, productivity law) themselves. We prefer to not remove this visualisation from the revised manuscript as it provides the visualisation of the effect in differences in ETAS parameters shown in Table 1. However, we recognize that the residual plot provides a clearer visualisation of the model fit and is a common visualisation in literature and add it in the revised manuscript.
As addressed by major changes listed above, the visualisation is now shown in Figure 4, introduced in Sect. 3.3 and discussed in Sect. 4.2.

*- In the section "Discussion of the model fit", many results sound trivial and can be easily explained by the well-known correlation among parameters. I would suggest making clearer what are the results that are new and cannot be explained by what we already know.*

We address this point in the revised version of the manuscript by modifying Sect. 4.1.

*- The caption of Figure 3 should contain an explanation of the colors used in the cells of the grid.*

We adjust the caption of said Figure in the revised version of the manuscript. This figure contains a legend indicating the meaning of the colours now, and the colours themselves are also hopefully more easily intuitively interpretable now.

*- One of the most interesting result is that the version of the ETAS model with alpha=beta is less performing producing "explosive" earthquake sequences (branching ratios larger than 1). I am wondering if the authors are using some maximum magnitudes (or corner magnitudes as well) in their simulation. As far as I know this is an outcome of ESHM20, and it could reduce drastically the problem of "explosive" sequences. A recent paper by Mancini and Marzocchi (2023) uses alpha=beta without having problems with "explosive" sequences. Maybe a few explanations on why the authors get explosive earthquake sequences could be worthwhile.*

In Sect. 3.3., we state that '*for all models, the maximum magnitude during the simulation phase is set to $m_{max} = 10.0$, which, due to the binning value of $\Delta m = 0.2$ corresponds to $m_{max} = 10.1$*'. While applying more detailed assessments of maximum magnitudes per region provided by ESHM20 might help with the explosive behaviour, we believe most of the effect is due to the tapered exponential temporal kernel. In ETAS$_{USGS}$, $\alpha = \beta$ with no taper in temporal kernel, and simulations not only converge, but underestimate the number of events in retrospective CSEP tests, due to a significant portion of aftershocks occurring after the observed period - here, even though the total branching ratio is higher than 1, the effective branching ratio for a 30-year period is lower. We clarify this point further and add the comparison of temporal kernels with Mancini and Marzocchi (2023) in the revised version of the manuscript.

**Reply on RC2**

*First, my apologies for the delay of this review!*

*This paper presents preliminary candidates for operational earthquake forecasting in Europe, based on variants of the well-established Epidemic Type Aftershock Sequence (ETAS) model. The authors draw on available harmonized datasets and a background model from the European Seismic Hazard model ESHM2020 to generate the pan-European models, making good use of prior painstaking work to collate the myriad of different national catalogs and methods. The study demonstrates convincingly that the ETAS models forecast some of the observed space-time clustering and may therefore be useful for real-time deployment. The detailed model evaluations include in-sample tests, which reveal some issues in some of the models as well as some issues with the applied tests, and they include a more severe out-of-sample pseudo-prospective test, and some sober and honest analysis of the relative performance of the models. They conclude that indeed 1-2 of the proposed ETAS models perform best and could serve as initial candidates (to be improved in the future).*

*Please see my (many) comments below. It's a well written and structured paper – and I have only a couple of major comments, everything else is largely around presentation and suggestions for improvements. I have one request to assess a forecast/simulation choice in more detail, and another to pursue some more analysis in the evaluation.*

We are grateful to the reviewer for the time invested in reviewing the study in a detailed and constructive manner and for providing insightful suggestions to improve the manuscript. We acknowledge that steps outlining both the major and minor changes to the manuscript are needed to address the current shortcomings.

*What is the influence of the chosen minimum probability level in histograms where no simulations have filled bins? How many number bins of the full space-time (and magnitude?) bins are set to this level? How frequently are they "observed"? How frequently do these empty number bins sit next to filled bins, suggesting perhaps that interpolation would be a better approach?*

Thank you for this comment. Many bins are indeed empty – the idea behind Fig. 5 in the original manuscript was to show that total information gain is dominated by cells where events do occur rather than ones where the event count is zero. We perform an additional analysis of how the choice of how to deal with empty bins affects the results, and add the relevant details to the revised manuscript.

*The authors rightly identify some undesirable features in the retrospective hypothesis tests applied, which is a useful contribution. But that means they should pursue other/additional methods, e.g. comparing the cumulative number trajectories of observed and synthetic catalogs (see specific comment below), comparing*

*(at least visually) the spatial distribution of the observed and synthetic catalogs, etc. Similarly, Figure 5 of the pseudo-prospective information gains don't seem very insightful – this is a first step to identify what's gone wrong/right, but would really benefit from some analysis, e.g. are the aftershocks not well captured in space, time or both? Why are the ETAS models sometimes performing worse than the ESHM2020 model? Even qualitative visual analysis would be useful.*

In the revised manuscript, we address this point in two steps: adding a plot comparing the cumulative number of events in Sect. 4 (Results), and performing more analyses to get a better understanding of the models' performance comparison in space and time, the methodology of which we add to Sect. 3 and the output to Sect. 4.

*Otherwise, please consider the below comments as hopefully constructive suggestions.*

**Abstract**

*L2 – is the purpose an aftershock forecast model? It sounds like it based on the first sentence. Please clarify.*

*The abstract would benefit from a sentence on the data the model is fit to, and any issues with "the" European catalog. Ie – what's harmonized?*

*It would help to clarify that these are currently 2D seismicity forecasts, not extended ruptures with depth (which might be fruitful areas for the future).*

*A clearer statement of the different models explored and their rankings would be of interest.*

*L10 How do the findings highlight these promising areas (last sentence)? Can you express it concisely or are these speculations or a wish list?*

The goal is to develop a 'full' earthquake forecasting model, i.e., one that produces forecasts continuously. With this sentence, we meant to express the fact that the main component of clustering is due to aftershocks, but we change this sentence to avoid confusion. We amend the abstract as allowed by the word limit (that we are already close to) to address the listed points.

**Introduction**

*L60 – missing a word after $m_c$ – maybe "more"?*

*The introduction could probably be shortened – e.g. the background to $m_c$ and the GR law etc seems more relevant for a thesis than a paper – unless the authors are specifically trying to reach a broader audience.*

*L75: As I understand it, the Mizrahi et al. (2021) model accounts for temporal – not spatial – variations in $m_c$. Is the model being extended? If so, mention it in the abstract and here.*

*L81: ESHM20 contains many models, some idea of how that's incorporated into ETAS in both the abstract as well as more discussion here would be appropriate.*

We address these points in the Introduction of the revised manuscript. The Mizrahi et al. (2021) model is based on the approximation that $m_c$ is locally constant and that observed seismicity at any point is representative of the unobserved seismicity below $m_c$. Although in their study, they only explore temporally varying $m_c$, no extension of the model is needed to apply it with space-time varying $m_c$.

We acknowledge that the introduction to GR and $m_c$ are basic terms, but would prefer to leave them in the manuscript as it helps to introduce the notation, e.g. since $a$ and $b$ are commonly used in literature, but the base-$e$ versions $\alpha$ and $\beta$ are less frequently encountered. Other points are addressed in the modified introduction.

*L112: These statistics of $m_c$ presumably come from Danciu et al. (2021)? Please clarify.*

Yes. We are adding the clarification in the revised version of the manuscript. We also add the visualisation in the supplementary materials, Figure S1.

*Figure 1: the red dots look black because of the marker edge colour. It's impossible to differentiate red from green dots in this figure. Try a bigger size or different/no edge colour.*

Thank you for pointing this out, we will amend the figure.

*L137: what do you mean here by "such differences"? Do you mean a difference in completeness magnitude, or a difference in the spatial locations? The second is indeed expected, the former perhaps less so. Please clarify.*

We were referring to differences both in composition methods and content of new catalogs. We recognize that changes in completeness would normally not be an expected occurrence, especially that it increases with time and rephrase this statement. However, given that catalog is composed of 'smaller' catalogs issued by local agencies, it is not unreasonable to expect at least one of the subregions will differ in properties compared to the training period at any given point in the testing period (especially in 'real-time' deployment when homogenized data may not be available).

*L140: Good to clarify. Does the binning require the discrete version of the b-value estimation procedure? Which one was used in Fig 1d? What are the uncertainties? Why show b=0.99? Perhaps this will be discussed below, then guide the reader.*

Yes, and we use this estimation procedure introduced in Tinti and Mulargia (1987). This is mentioned later, but we agree it should also be included when introducing the binning and the figure. The two $b$-values shown in the plot are one used by ETAS variants later, which we also explain later, but now also add at this point.

*L142: I would appreciate a figure of the long-term rates in these two models.*

The figure is added in the revised manuscript, either in the main manuscript or as supplementary material.
It is added as supplementary material.

*L146-153: this sounds like Methods, rather than data. I recommend more discussion of the two long-term models (ie a figure of the two models and a bit more background on both models). Then move the discussion of how you map these into the background rate of ETAS into a subsection in Methods, including some discussion of how this long-term seismicity might or might not be compatible with the background (independent) rate in an ETAS model would be appropriate (or else just extend this section).*

While this data requires some preprocessing, the procedure itself is not a method we develop, but a product of ESHM20 (Danciu et al., 2021) used here as input. We agree that the discussion would benefit from more details and we address this point in the revised manuscript.

*L187: again, if the ESHM20 considers long-term rate as the declustered rate, there is some consistency (even that's debatable), but if the ESHM20 rate is the long-term, i.e. average rate, then setting this to the background rate may lead to an overestimation of the total rate (instead the average model rate should equal the ESHM20 rate, see e.g. Field et al. (2017)). Or are you solely considering the relative spatial information, rather than the absolute rate? I think clarification of this point earlier in the paper would help. (You clarify this in L242, and I agree this is a good approach, but I'd mention it earlier.)*

We are indeed considering only the relative spatial information, exactly because it would lead to the overestimated background rate otherwise. As suggested, we clarify this earlier in the revised version.

*L193: ETAS$_{USGS}$: A few more details would help – isn't this an aftershock-only model? Are these truly one-size fits all for the entire globe or are they regionalized?*

The true USGS software uses parameters specific to tectonic regimes, which were obtained by fitting global data from the respective tectonic regimes, but they also have a set of parameters representing the global average, which we employ for aftershock modelling, while using 'our' background seismicity inferred by ETAS$_0$. We add this additional explanation to the revised manuscript.

*L203: burn period → burn-in period*

Thank you for noticing - corrected.

*L213: which events' locations exactly – the new/simulated background events, or do you not include all background events in the burn-in period?*

The locations of simulated background events in the period we are simulating (the burn-in period is only used to simulate aftershocks of the events from that period that occur in the period we are simulating). Clarification added in the sentence.

*L258: It would help to a bit more precise here, and to consider the role of the maximum magnitude or a tapered GR law. "Exploding" aftershock behavior may occur with some finite probability when the branching ratio r is > 1, and sequences are finite with probability 1 when r = 1. The calculation of r involves the GR law, and thus depends on your choice of pure GR vs tapered/cut GR, and thus in the latter case on Mcorner/Mmax. First, please include a short discussion/justification on your choice earlier when you introduce the magnitude distribution (including the potential for spatial variations). Second, in the context of constraining parameters for subcritical branching, these modifications matter and thus the parameter constraints change. The branching ratio r may well be < 1 if the GR is tapered/cut while it may not with a pure GR law. Sornette and Werner (2005) derived r equations for a truncated GR. Finally, clarify what 'e' is (another variable or Euler's number?).*

We take into account the maximum magnitude when calculating the branching ratio in cases when $\alpha - \rho\gamma > \beta$. Using more detailed assessments of maximum magnitudes per region and spatial variations in magnitude distributions in general provided by ESHM20 might help with the explosive behaviour, however, we assign most of this effect to the tapered exponential temporal kernel. In ETAS$_{\text{USGS}}$, we set $\alpha = \beta$ with no taper in temporal kernel, and simulations not only converge, but underestimate the number of events in retrospective CSEP tests, due to a significant portion of aftershocks occurring after the observed period, resulting in a lower 'effective' branching ratio lower than the total one higher than 1. We clarify this point further in the revised version of the manuscript. We address other points as well and will add details about this to the revised manuscript.

*L264: "standard" in the USGS software?*

The formulation we are referring to is indeed one used by USGS, by 'standard' formulation we mean one that uses $(t+c)^{-p}$ rather than $(t+c)^{-(1+\omega)}$ (and similar for spatial and productivity) that we use, which is a bit less common in literature. We rephrase for clarity.

*L286: Does this mean that the explicitly spatially-variable models forecast spatially uniform background rates in each spatial ESHM zone? I don't believe so, but please clarify – if I understand correctly, the $ETAS_{bg}$ models use the same procedure to simulate background events as the $ETAS_0$ model, but both the probability of being a background rate has changed because of different parameters & the additional constraint that the rates in different zones are relatively constrained. Is that correct?*

The procedure is different, the models with spatially-varying ESHM-informed background rate do simulate locations uniformly within each grid cell (0.1°lat × 0.1°lon), the total number of background events is distributed according to the ETAS inversion output, but their relative locations are enforced to follow the ESHM input. We clarify this in the revised manuscript.

*L292: Ah, so it is a truncated GR law – does this influence the branching ratio constraint alpha = e? See earlier comment.*

We mention the truncated GR law earlier in the article when the previous question arose.

*L295: "true" → observed? Here and below.*

Yes, clarified in the revised manuscript.

*L320: Why only focus on retrospective consistency testing? 7 years of daily out-of-sample testing seems like a great start, even if not 25 years. I recommend doing these or similar tests, or some of them – the information gain assess a different aspect of the models, but tell us less about how close to the data the models are (and while the non-Poisson cell-wise LL scores are a great improvement, they still neglect known correlations between spatial cells).*

Thank you for this suggestion. In response to other comments, also of the other reviewer, we perform additional analyses both for the retrospective and pseudo-prospective tests and add the relevant findings to the revised manuscript.

*L338: What is the influence of this implementation and the particular choice of a minimum waterlevel on the overall scores?: how many zero-bins are there in a 100k simulation set per day? How does the waterlevel compare with the background rate (and its Poisson process implied probabilities)? How "rough" is the zero-bin distribution, ie could an interpolation procedure approximate the ETAS forecasts better than this water-level?*

We add this analysis in the revised manuscript, see also our response to the previous (major) comment.

*L419: It would be illustrative to give the range of values in days or years after which the Omori law is significantly tapered in this formulation (ie give the range of tau in days). (I see the range in L436, but give units in years).*

The ranges are given in more interpretable units in the revised version of the manuscript.

*Parameters: Can you connect your discussion of the ETAS parameter estimation with the literature more directly (e.g. do you see similar patterns to Seif et al. (2017), and others...)?*

We extend and clarify the parameter inversion output discussion in the revised manuscript.

*Do you have LL scores for your various ETAS model fits? Can you compare the in-sample fit using LL and AIC? It's a useful indicator of fit, though not necessarily about predictive skill.*

We calculate these values and will add them to the manuscript if it turns out to add valuable information to the study.

*Figure 3: Please clarify these are simulations over the entire training period. Also, L279 states 100k catalogs were simulated, but here it states only 10k. I would clarify the timeframe of the retrospective testing also in the relevant Methods section around L279. These are not next-day forecasts, as in the pseudo-prospective testing section.*

Thank you for spotting the inconsistency - the suggested changes and corrections are done in the revised version.

*Figure 3a: I would recommend looking at the cumulative counts in each of the catalogs and compare with the observed count. You could look at the cumulative 95% model range and compare with the observed number. But you also get important insights into how well the model is qualitatively and quantitatively reproducing the features you are hoping it will, namely aftershock clustering and background rate.*
*Fig3: As part of these formal tests, I would also look at visual fits, e.g. Fig 1d seems to show a comparison of observed data and the magnitude forecasts from the different b-value models. I know the formal tests do this comparison quantitatively, but I would discuss the visual fit (which quite clearly favours the higher b-value).*

We are adding the figure showing cumulative event counts comparison in the revised manuscript. We also acknowledge the need to compare the magnitude distributions visually and agree on the assessment that the higher *b*-value estimate looks more correct.

*Supercritical branching when alpha is fixed: these branching ratios are indeed very high, but supercritical has been found before, e.g. Seif et al. (2017) and some of the other references you used have found similar issues. There's clearly model misspecification that's driving this bias, so it would be interesting to discuss possibilities: changing background rates, anisotropic spatial aftershock triggering, spatial variation in incompleteness?*

Thank you for the suggestion - we are extending this discussion.

*L476: "suspiciously well" in spatial terms means that the models are too smooth, ie that the events occur too close to likelihood peaks without the scatter expected if the model were the data generator. That should be spelt out.*

We agree and amend the phrasing.

*L479: refer to Fig 1d in this discussion of magnitude distribution fit to data.*

We agree this is an additional argument for the $b$-positive ETAS variants and add it to the discussion.

*L481: is it true that "$ETAS_{bg}{}^{b+}$ uses the same spatial distribution for placing the background events" as $ETAS_0{}^{b+}$? As commented above, I understood two differences: the relative rates between zones are constrained, and the absolute rate is different, and the parameters are different, so the probability of being a background event is also affected. Could you clarify (here and above where commented)?*
*Irrespectively, and again, I would visually compare the forecast and observed magnitude distributions and use this to support your case that the difference in magnitude forecast performance of the two models is indeed a surprising result and indicates an issue with the M-test (and the S-test, although I'm not sure I understand your detailed argument here, see above).*

They indeed don't use the same spatial distribution of background events. This paragraph is corrected in the revision. We add the magnitude distribution comparisons to the discussion as mentioned in earlier comments, too.

*L482: "known issue": as you know, Francesco Serafini and others identified a correlation between the M-test and the N-test, which is indeed problematic and being fixed. You might cite Francesco Serafini (personal communication, 2024), or perhaps refer to the manuscript in prep that might be submitted by the time this is published.*
*Given the (reasonable) caution you advise in interpreting these M/S/PL results, you might use visual checks, including of the spatial distributions, appropriately averaged (or not) over the many simulations, to show the extent to which models reproduce the features you expect them to.*

We agree that visualizations would help assess the spatial and magnitude distributions. As described in our responses to previous comments, we will perform these additional analyses and include relevant findings either in the main manuscript or as supplementary material. We also agree that the study being conducted by Serafini et al. is highly relevant to this discussion and cite it in the revised manuscript.

*Figure 4a would benefit from a panel underneath with the magnitude-time plot of the seismicity, which will help identify clustering and quieter periods and visually link them to the likelihood gains (or losses).*

*Figure 4b/c: these tables are hard to digest, despite the nice colors. Could you please plot (instead or in addition) a figure instead showing mean information gain over the ESHM20 model with 95% range (which I believe you get from the paired t-test?) to indicate significance?*

We recognize these suggestions could improve the readability of the figure and will try out alternative visualisations for the updated version of the manuscript.

This figure is now modified to include the seismicity through time, and information gain is also shown for no spatial binning. Matrices showing the MIG and significance are hopefully now more easily understandable. The requested figure showing confidence intervals for MIG (one-sided t-test) for $0.1° \times 0.1°$ spatial binning (left) and no spatial binning (right) is attached here, but we preferred the tabular view in Figure 6 as it shows pairwise significance, rather than only in comparison to ESHM20. The pairwise view is commonly done and this particular visualisation is done by Iturrieta et al. (2024).

[Figure]

[Figure]

*L506: "achieve a significance level below 0.05" is I believe not the right interpretation of these tests – you might just state that their p-values are below the critical value.*

True, we rephrase this statement.

*L515: "could be expected" – yes, of course, but the purpose is to develop a model that does indeed use this knowledge and puts this into practice. It might be a basic statement for some of the community, but not for others and potential users. I would emphasise the success in finding one or several models that*

*do show substantial improvement in predictive skill, and that this skill shows up during periods of clustering (presumably, although it would be good to illustrate/visualize this explicitly, as suggested eg via the modified figure 4a that links cumulative LL scores trajectories to seismicity magnitude-time plots to show where the clustering occurs). And I would be careful with absolute statements like "poor performance" – this is relative to the other ETAS models as measured by your metric, which is (a) an approximation of the correlation of seismicity between spatial bins and (b) requires many model simulations, which may not be sufficient to fill all "bins" with simulations (see waterlevel comment). So I'd be a bit more careful and emphasise this first milestone – a European ETAS model that does what it says on the tin: forecast aftershocks and capture some of the time dependence of seismicity, clearly much better than a time-independent model. And yes, there are some unexpected results here, which would be good to understand in more depth in order to improve on this first attempt.*

We appreciate this insightful summary of the output of the study and rephrase the conclusion according to the given suggestions.

*With respect to the spatial LL comparison in Fig 5, I suspect there are patterns, and some more analysis would be required to understand these patterns. It's surprising to see such strongly negative gains between an ETAS model and the ESHM20 model, for instance. Are these individual events? Clusters gone awry? And what's going on at the mid-oceanic ridge? This section is a bit thin, so at least some more discussion would help. What explains the major differences in some cells between the two ETAS versions? Or at least what happens there?*

We will look into this in more detail and provide the relevant outcomes in this section in the revised manuscript.

As the idea behind this figure was to look into spatial patterns on regional scale, we prefer to leave it as is, with the modification that it now shows $ETAS_{bg,\alpha}{}^{b+}$ instead of $ETAS_{bg}$, since it is arguably the most promising alternative with regards to overall testing scores, leaving $ETAS_0$ which achieves highest MIG now. The difference between models is addressed with a different sensitivity study focusing on spatial binning, showing that the order of models in pseudo-prospective testing changes substantially when spatial component is disregarded.

**References**

Danciu, L., Nandan, S., Reyes, C. G., Basili, R., Weatherill, G., Beauval, C., Rovida, A., Vilanova, S., Sesetyan, K., & Bard, P.-Y. (2021). The 2020 update of the European Seismic Hazard Model-ESHM20: Model Overview [Publisher: ETH Zurich]. *EFEHR Technical Report*, *1*.

Field, E. H., Milner, K. R., Hardebeck, J. L., Page, M. T., van der Elst, N., Jordan, T. H., Michael, A. J., Shaw, B. E., & Werner, M. J. (2017). A spatiotemporal clustering model for the third Uniform California Earthquake Rupture Forecast (UCERF3-ETAS): Toward an operational earthquake forecast. *Bulletin of the Seismological Society of America*, *107*(3), 1049–1081.

Iturrieta, P., Bayona, J. A., Werner, M. J., Schorlemmer, D., Taroni, M., Falcone, G., Cotton, F., Khawaja, A. M., Savran, W. H., & Marzocchi, W. (2024). Evaluation of a Decade-Long Prospective Earthquake Forecasting Experiment in Italy. *Seismological Research Letters*. https://doi.org/10.1785/0220230247

Jordan, T. H., Chen, Y.-T., Gasparini, P., Madariaga, R., Main, I., Marzocchi, W., Papadopoulos, G., Yamaoka, K., & Zschau, J. (2011). Operational Earthquake Forecasting: State of Knowledge and Guidelines for Implementation. *Annals of Geophysics*.

Mancini, S., & Marzocchi, W. (2023). SimplETAS: A Benchmark Earthquake Forecasting Model Suitable for Operational Purposes and Seismic Hazard Analysis. *Seismological Research Letters*, *95*(1), 38–49. https://doi.org/10.1785/0220230199

Mizrahi, L., Nandan, S., & Wiemer, S. (2021). Embracing data incompleteness for better earthquake forecasting [Publisher: Wiley Online Library]. *Journal of Geophysical Research: Solid Earth*, *126*(12), e2021JB022379.

Seif, S., Mignan, A., Zechar, J. D., Werner, M. J., & Wiemer, S. (2017). Estimating ETAS: The effects of truncation, missing data, and model assumptions. *Journal of Geophysical Research: Solid Earth*, *122*(1), 449–469. https://doi.org/10.1002/2016JB012809

Serafini, F., Naylor, M., Bayliss, K., Werner, M., Iturrieta, P., Bayona, J. A., Mizrahi, L., & Han, M. (2024). Comparing consistency tests for magnitude distributions [in preparation, personal communication].

Sornette, D., & Werner, M. J. (2005). Apparent clustering and apparent background earthquakes biased by undetected seismicity. *Journal of Geophysical Research: Solid Earth*, *110*(B9), 2005JB003621. https://doi.org/10.1029/2005JB003621

Tinti, S., & Mulargia, F. (1987). Confidence intervals of b values for grouped magnitudes. *Bulletin of the Seismological Society of America*, *77*(6), 2125–2134. https://doi.org/10.1785/BSSA0770062125

van der Elst, N. J., Hardebeck, J. L., Michael, A. J., McBride, S., & Vanacore, E. (2022). Prospective and retrospective evaluation of the U.S. Geological Survey public aftershock forecast for the 2019-2021 Southwest Puerto Rico Earthquake and aftershocks. *Seismological Research Letters*, *93*(2A), 620640. https://doi.org/10.1785/0220210222

---

## Author Response (AR2)

**Towards a Harmonized Operational Earthquake Forecasting Model for Europe: Author's Response**

Marta Han[1], Leila Mizrahi[1], and Stefan Wiemer[1]

[1]Swiss Seismological Service (SED), ETH Zurich

**Report #1**

*The revised version is much improved. Below I have some remaining suggestings to improve presentation and clarity. And I do have some queries and comments about the water-level results, specifically the sensitivity of the information gain trends with respect to what seem minor changes in the water-level, which are very surprising and need some technical checks and some interpretation (not a solution). The sensitivity may be reduced by using a more appropriate Poisson baseline (see relevant comment below) or using the water-level only when quakes do appear in zero- forecasts bins (and not in all zero-forecast bins). After that, I can recommend publication.*

We thank the reviewer for another round of helpful insights, pointing out the dependency on this parameter in the first place, and further suggestions to help address the water-level sensitivity in our results. We hope that our newly revised manuscript addresses the issue more appropriately.

*Figure 1: b-value estimates should include uncertainties estimates.*
*Fig1: The difference between b and b+ seems large, and here b+ seems to give larger values than usual. Which delta m (ie equivalent mc) did you use for b+? Is it possible that b+ is biased for this very particular dataset? It just seems larger than expected.*

We add the uncertainties to the revised manuscript and specifically to the caption of Fig. 1. Both $b$-value estimates take into account the magnitudes' binning of $\Delta m = 0.2$ and apply the correction introduced in Tinti and Mulargia (1987) to avoid biases. Completeness is spatially and temporally varying, and differences $m - m_c(x, y, t)$ considered in place of pure magnitudes to bring the mixed distribution closer to GR, and completeness of $m - m_c(x, y, t)$ (for $b$-positive) is 0.2. Since the region considered in our study is very large, the key

underlying assumption behind applying the b-positive method, namely that after an event of a certain magnitude, any event larger by a certain increment will be detected, might not apply. However, the results obtained in our experiments suggest that the b-positive estimator yields a more appropriate magnitude distribution than the classical estimator. This suggests that if one estimator is biased, it might be the classical one. However, overall, one of our main conclusions remains that using a single $b$-value for a large and diverse region such as Europe may be an oversimplification of the problem. The estimation of $b$-values should be improved by reassessing the completeness estimates and using a spatially (and maybe even temporally) varying $b$-values and we plan to work on this in the future.

*L203: "no substantial evidence" I think requires some references, even if it's your interpretation of some results that do include such updating.*

We agree and change the formulation of the statement, now stating that there is no agreed-upon technique to update models to specific sequences. We also add relevant literature, specifically describing the different model updating strategies of Italy, New Zealand, and the United states.

*L210: clearly state here that $ETAS_0$ contains one spatially uniform background rate.*

We add the clarification to the model specification. However, we would also like to emphasise that while the background rate is uniform during the inversion of parameters, when producing forecasts, background events are simulated at locations where events were observed in the training catalog, weighted by their background probability estimated during the inversion procedure. Hence, in the forecast that is the final output of every model, background events are not located uniformly in space (see Fig. 5).

*L276: "due to their under-representation in training data" – There are multiple papers suggesting it's the anisotropy of the aftershocks compared to the isotropic model (Hainzl et al., 2008; Helmstetter et al., 2005; Zhang et al., 2020), as well as the covariance between K and alpha in the likelihood function (Sornette and Werner, 2005 and probably others)*

We thank the reviewer for this feasible explanation and add it along with listed literature to the revised manuscript.

*L308: pls cite these papers in support of the pyCSEP toolkit efforts:*

- *Savran et al., (2022). pyCSEP: A Python Toolkit For Earthquake Forecast Developers. Journal of Open Source Software, 7(69), 3658, https://doi.org/10.21105/joss.03658*

- *Savran, W. H., Bayona, J. A., Iturrieta, P., Asim, K. M., Bao, H., Bayliss, K., ... & Werner, M. J. (2022). pyCSEP: a Python toolkit for earthquake forecast developers. Seismological Society of America, 93(5), 2858-2870.*

*Or alternatively/additionally: L496: pls consider adding the JOSS citation here to the SRL citation. The former is the peer-reviewed code base and associated online documentation, while the latter describes the software and motivation.*

We agree that these citations are highly relevant to the topic and acknowledge the tremendous efforts done by CSEP community. Thus, we add these citations in the relevant parts of the manuscript.

*L383: Actually, the log score is well defined when the forecast is zero and the observed count is zero: the likelihood is exactly 1 and the log likelihood is zero. Secondly, the log score is still well defined, it is negative infinity, when the forecast is zero and there are indeed events. (see comment at the end of the review)*

We amend the phrasing in the revised manuscript to avoid confusion.

*L393: Could you clarify how the two year period helped set the value? Also, which benchmark model are you referring to (Poisson – as mentioned below)?*

Using the 2-year validation set, we inspected plots similar to Fig. S4 to identify water levels for which models perform worse and then verified that it is because either the water level is too low, penalizing its usage heavily, or too high, causing ETAS models to score lower in bins where they forecasted events, but still scoring lower also in bins where water level is used (newly added Fig. S5 in response to the last comment shows this for high water levels). The benchmark model is indeed the time-independent Poisson one (with ESHM seismicity rates per spatial bin). We clarify the statement in the revised manuscript.

*L498: true → observed*

Corrected here and in a number of other places in the manuscript.

*L504: for clarity state why you exclude $ETAS_\alpha$ and $ETAS_{bg,\alpha}$ here (as you do in the caption of Fig 4).*

We address this point in the revised manuscript.

*Figure 5: Are the white cells in these figures those where no events were simulated? What are the units on the colour scale (expected events per year)? Do these figures suggest a water level based on the background?*

The events are counted in total during the entire 25-year training period. In the white cells, indeed no events have been simulated. We extend the caption of said figure to include this information. No water level is used to create these figures. They could, however, be used to create a background-based water level.

*Figure 6: I'm quite surprised by these results. Why isn't there a stronger step change at the time of larger quakes, e.g. the M7 in late 2020? It's so evident in late 2018 for a smaller mainshock, but the other large quakes don't seem to generate much information gain for the ETAS models. It's curious – perhaps few aftershocks above the completeness? Please provide a short interpretation.*

We add a short analysis in the revised manuscript. Smaller number of aftershocks above $m_c$ seems to be a plausible explanation, as well as the fact that forecasts are issued at midnight. If an event occurs relatively shortly after midnight, the most productive period is not part of the next testing day.

*Is the Poisson model uniform or spatially variable? Is the Poisson rate the average rate over the pseudo-prospective period (which would give it more information than the ETAS models got) or over the retrospective training period (which seems fairer, and one I'd recommend)? Please clarify in the text.*

*In addition, I wonder whether you want to exclude the models you've discounted based on the retrospective tests in Figure 6? You show that some of these excluded models have the highest information gains, but then discount them based on retrospective tests. Is it still useful to show them? In any case - make sure these two sections are very consistent with each other.*

The Poisson model is spatially variable, with the rate in each cell being the long-term seismicity rate (normalised from annual to daily) from ESHM20, which includes no information from the pseudo-prospective experiment period, but includes seismicity up to 2015, and additional information such as physical tectonic properties and historic seismicity information. We clarify this better in the revised manuscript in Section 3.2.

Regarding the discounted models, we amend the discussions to be more consistent, but would prefer to keep all models in Figure 6. Better performance of models that fail retrospective tests is still valuable information, allowing comparison between (a) and (b) parts of Figure 6, and our conclusion is that we prefer models used for OEF pass the consistency tests, not that models that fail them should not be disregarded altogether (and it should be investigated further why some are better with shorter time windows or smaller spatial cells).

*L628: The water level only needs to be invoked when you have observed quakes but the forecast is zero. If there are zero quakes, the forecast is technically correct and should give probability 1, i.e. log score zero, ie there is no log score penalty (a perfect prediction). So the water level should only be applied where you do see events but the forecast is zero. Is this how you've implemented it?*

*Figure S4 is quite surprising! Slight changes in water level generate substantial changes in overall trends, and even rankings are affected. And the trends (ie overall positive or negative against Poisson) seem to change randomly even for small changes in water level, which is surprising if the baseline stays the same. Did you maintain the simulated forecasts between these plots and only changed the water-level or could there be an effect due to different simulated forecasts here too? To isolate the water-level effect, I think you should keep the simulated forecasts the same, to make it's not differences in stochastic simulations that generate these differences.*

*More importantly, how do you explain that models perform worse than Poisson in panel top-left when the water level is relatively high (but still less than Poisson?), but then better when the water-level is halved (second panel on the left), then worse again when divided by another 100? It'd be good to label the panels for this discussion.*

*My recommendation is to check the technical details above (fix simulations between different water-levels; only use it when there are quakes; explain the reversal of trends if it persists). It's interesting to point out this sensitivity, because it helps the community develop better methods (hopefully). You don't need to solve it here.*

The simulated dataset is kept fixed already, only the procedure producing the forecasts based on those simulations is adapted to each water level. In the revised manuscript, we specify this information, and add the plot analog to S4 in supplementary materials for when water level is invoked only when needed. The comparison allows for further conclusions in this section answering some of the important questions raised here. In the main plot, we still opt for the version where water level is always distributed over all bins. Namely, our goal is to produce forecasts near real-time and test them truly prospectively; in that case, we cannot know if water level will be needed and distribute it accordingly and therefore, we believe that the current experiment setting better reflects the conditions of truly prospective testing.

**Report #2**

*The authors did a good job in addressing satisfactorily my main comments. I am not still sure to agree with some statements reported in this paper, and with some modeling choices. But, overall, I do think that, in this form, the manuscript can stimulate further research and additional thoughts on this important problem.*

We thank the reviewer for the positive feedback and constructive suggestions in the first round of reviews. The efforts described in the paper are an ongoing project and we hope to address some of the worries in our future work.

**References**

Hainzl, S., Christophersen, A., & Enescu, B. (2008). Impact of Earthquake Rupture Extensions on Parameter Estimations of Point-Process Models. *Bulletin of the Seismological Society of America*, *98*(4), 2066–2072. https://doi.org/10.1785/0120070256

Helmstetter, A., Kagan, Y. Y., & Jackson, D. D. (2005). Importance of small earthquakes for stress transfers and earthquake triggering. *Journal of Geophysical Research: Solid Earth*, *110*(B5). https://doi.org/10.1029/2004JB003286

Sornette, D., & Werner, M. J. (2005). Constraints on the size of the smallest triggering earthquake from the epidemic-type aftershock sequence model, Båth's law, and observed aftershock sequences. *Journal of Geophysical Research: Solid Earth*, *110*(B8), 2004JB003535. https://doi.org/10.1029/2004JB003535

Tinti, S., & Mulargia, F. (1987). Confidence intervals of b values for grouped magnitudes. *Bulletin of the Seismological Society of America*, *77*(6), 2125–2134. https://doi.org/10.1785/BSSA0770062125

Zhang, L., Werner, M. J., & Goda, K. (2020). Variability of ETAS Parameters in Global Subduction Zones and Applications to Mainshock–Aftershock Hazard Assessment. *Bulletin of the Seismological Society of America*, *110*(1), 191–212. https://doi.org/10.1785/0120190121